# A molecularly defined basalo-prefrontal-thalamic circuit regulates sensory and affective dimensions of pain in male mice

Guoguang Xie [1,2,3], Yiqiong Liu[1,2,3], Xuetao Qi [1,2,3], Aritra Bhattacherjee[1,2,3], Chao Zhang [1,2,3] & Yi Zhang [1,2,3,4,5] ✉

Both the medial prefrontal cortex (mPFC) and thalamus have been implicated in pain regulation. However, the roles of the mPFC-thalamus connection in pain and how the mPFC modulates nociceptive processing remain unclear. Here, we show that the mPFC neurons projecting to thalamus, marked by *Foxp2* expression, are deactivated in both acute and chronic pain in male mice. Persistent inactivation of the mPFC *Foxp2*+ neurons enhances nociceptive sensitivity, while their activation alleviates multiple aspects of pain. Circuit-specific manipulations revealed that the projections to parataenial nucleus, mediodorsal and ventromedial thalamus differentially modulate sensory and affective pain. Additionally, the mPFC *Foxp2*+ neurons receive cholinergic input from the basal forebrain, particularly the horizontal diagonal band (HDB). Notably, activation of the α4β2-containing nicotinic acetylcholine receptor in mPFC exerts antinociceptive effects in *Foxp2*+ neuron-dependent manner. Together, our study defines an HDB→mPFC^*Foxp2*→thalamus circuit essential for sensory and affective pain modulation and underscores the therapeutic potential of targeting mPFC cholinergic signaling in chronic pain management.

Pain is an unpleasant sensory and emotional experience associated with actual or potential tissue damage and is essential to protect the body from potentially harmful stimuli[1,2]. Chronic pain, however, is not just a temporal continuum of acute pain but one of the major causes of human suffering, affecting about 20% of the global population with immense clinical and economic consequences[3,4]. Unfortunately, current pain management strategies are largely unsatisfactory, with inadequate pain relief and multiple adverse effects[5,6]. Thus, understanding the mechanisms underlying pain processing remains an urgent need for developing effective treatments.

Accumulating evidence supports the crucial roles of medial prefrontal cortex (mPFC) in regulating multiple aspects of pain, including the sensory, affective, and cognitive dimensions[7–11]. Neuroimaging studies in humans have shown functional deactivation and altered connectivity of mPFC in patients with chronic pain[12–14]. Consistently, preclinical studies have also revealed the diminished activities of mPFC in several forms of chronic pain and the critical roles of mPFC in pain relief[15–18]. The mPFC is widely connected with other brain regions, including the periaqueductal gray (PAG), thalamus, amygdala, and basal nuclei[19]. As part of the descending modulation pathway that governs the noxious inputs in the spinal cord, the mPFC-to-PAG circuit in pain processing has been well documented[5,16,17]. However, it is much less clear about the roles of the mPFC outputs to other brain regions in pain regulation, as well as how the mPFC modulates the processing of noxious information in the brain.

The thalamus, a major target of the mPFC output, serves as a key hub for integrating and processing nociceptive information ascending from the spinal cord and modulates both sensory and affective

[1]Howard Hughes Medical Institute, Boston Children's Hospital, Boston, MA, USA. [2]Program in Cellular and Molecular Medicine, Boston Children's Hospital, Boston, MA, USA. [3]Division of Hematology/Oncology, Department of Pediatrics, Boston Children's Hospital, Boston, MA, USA. [4]Department of Genetics, Harvard Medical School, Boston, MA, USA. [5]Harvard Stem Cell Institute, Boston, MA, USA. ✉e-mail: yzhang@genetics.med.harvard.edu

dimensions of pain through the lateral and medial pathways[5,20]. Imaging studies have linked chronic pain to thalamic alterations, including reduced volume, activity, and imbalanced transmitters[12,14,21]. Recent findings also highlight the critical functions of the connections between the thalamus and other cortices, such as the motor cortex and ACC, in regulating affective pain[22,23]. Despite the strong anatomical connections between mPFC and thalamus and their critical roles in pain and other behaviors such as attention and cognition[24,25], the specific functions of the mPFC-to-thalamus circuit in pain processing remain largely unexplored.

The mPFC receives inputs from multiple brain regions[19]. Among these, the strengthened glutamatergic input from basolateral amygdala (BLA), which synapses onto the inhibitory neurons in mPFC and sends feedforward inhibition to the pyramidal neurons, has been reported as a key mechanism underlying the deactivation of mPFC in chronic pain[17,18]. However, the roles of other mPFC inputs in chronic pain remain largely unknown. Notably, the mPFC receives dense cholinergic innervation from the basal forebrain, a pathway involved in various functions such as learning, attention, and arousal[26]. Although the clinical importance of cholinergic signaling in pain has been well recognized[27], the roles and the circuit mechanism of cholinergic modulation of mPFC in pain are not fully understood. Moreover, the mPFC is highly heterogeneous, with differential structural and functional dysregulation of the mPFC in chronic pain[8]. The variable functional responses, either pain relief or exacerbation following mPFC activation[8,28,29], might result from the cell heterogeneity across subregions and layers of mPFC, which requires higher cell-type and projection-specific resolution to accurately delineate.

In this study, we identify *Foxp2* as a marker of the mPFC neurons that project to the thalamus and show that these neurons are deactivated under acute and chronic pain. Using cell-type-specific and circuit-specific manipulations, we demonstrate that activation of the mPFC *Foxp2*+ neurons alleviates both the sensory and affective components of pain by targeting distinct thalamic subregions. Additionally, we show that the mPFC *Foxp2*+ neurons receive cholinergic innervation from the basal forebrain, particularly the horizontal diagonal band of Broca (HDB), and that activation of the cholinergic projection is antinociceptive. Notably, we found that activation of the α4β2-containing nicotinic acetylcholine receptor (nAChR) in mPFC relieves pain, which requires the *Foxp2*+ neuronal activity. Collectively, our study reveals a mechanism by which the mPFC regulates pain through projection to the thalamus, and highlights mPFC cholinergic signaling as a potential therapeutic target for chronic pain treatment.

## Results

### *Foxp2* specifically marks the mPFC outputs to the thalamus

The cerebral cortex is organized into layers with diverse neuron subtypes that have distinct projection patterns distributed in a laminar pattern[30]. In the mPFC, the projections to the thalamus, a hub for pain processing, primarily originate from the layer (L) 6 corticothalamic (CT) neurons[31,32]. Surprisingly, single-molecule fluorescence in situ hybridization (smFISH) revealed that *Ntsr1*, a widely used L6 CT neuron marker in the cortex[22,33,34], is broadly expressed from L2 to L6 in the mPFC (Supplementary Fig. 1a), indicating the different organization of the mPFC compared to other parts of the cortex. Integrative analysis of our previous single-cell RNA sequencing and spatial transcriptomic data indicated *Foxp2* and *Oprk1*, respectively, mark the L6 CT and L6 intratelencephalic (IT) neurons[35,36], the two major excitatory neuron types in L6 (Supplementary Fig. 1b). This is consistent with previous studies suggesting exclusive expression of *Foxp2* in CT neurons of the sensory cortex[33,37–40].

To determine whether *Foxp2* can serve as a marker for L6 CT neurons in the mPFC of adult mice, we injected cholera toxin subunit B (CTB) 555, a retrograde tracer, into the medial thalamic nuclei and performed smFISH for *Foxp2* in the mPFC (Fig. 1a). We found that the CTB+ cells are mainly located in L6 with over 90% of them express

*Foxp2*, while around 90% of the *Foxp2*+ neurons are co-labeled with CTB (Fig. 1a), indicating that the *Foxp2*+ neurons account for the major outputs of mPFC to thalamus. To examine whether the mPFC *Foxp2*+ neurons project to other major mPFC targets, CTB was also injected into the PAG (Fig. 1b), contralateral mPFC (Supplementary Fig. 1c), amygdala (Supplementary Fig. 1d), or nucleus accumbens (NAc) (Supplementary Fig. 1e), respectively. Minimal colocalization of CTB and *Foxp2* was observed in these conditions (Fig. 1b and Supplementary Fig. 1c–e). These findings confirm that *Foxp2* specifically marks the major outputs of mPFC to the thalamus.

Given that the outputs of mPFC to the thalamus mainly originate from the L6 CT neurons, we then asked whether *Foxp2* specifically marks this distinct neuronal subtype. To this end, we performed smFISH for *Foxp2*, *Slc17a7* (a marker for glutamatergic neurons), and *Gad1* (a marker for inhibitory neurons), which revealed that over 95% of *Foxp2*+ cells express *Slc17a7* (Fig. 1c), indicating that *Foxp2*+ cells are predominantly glutamatergic neurons. To further characterize the distribution of *Foxp2*+ cells in the mPFC, smFISH for *Foxp2, Etv1* (a marker for L5), and *Ctgf* (a marker for L6b) was performed with the mPFC sections along the anterior-posterior axis (Fig. 1d and Supplementary Fig. 2). Quantification of the fluorescence signals confirmed the expression of *Foxp2* in L6, including L6b, with about 10% of *Foxp2*+ cells also expressing *Etv1* (Fig. 1d and Supplementary Fig. 2a–c). Further quantification of the fluorescence signals in the mPFC from pia to white matter revealed that the *Foxp2*+ cells are mainly located in L6 and to a much less extent in deep L5, especially in the anterior part of mPFC (Supplementary Fig. 2d). Since *Foxp2* and *Oprk1* are indicated to respectively mark the L6 CT and IT neurons (Supplementary Fig. 1b), we examined their expression in the mPFC and found that they label two distinct neuronal populations in L6 with minimal overlap (Fig. 1e). Because there are also a small number of *Foxp2*+ neurons in L5, where resides the L5 IT neurons, as well as the extratelencephalic (ET) neurons labeled by *Pou3f1*[35] and near-projecting (NP) neurons labeled by *Tshz2*[41], we then performed smFISH and found little overlap between *Foxp2* and *Pou3f1* (Fig. 1f) or *Tshz2* (Fig. 1g). Collectively, these results indicate that *Foxp2* labels a distinct neuronal subtype in L6 different from the IT, ET and NP neurons in the mPFC, consistent with the tracing results.

To further confirm that the mPFC *Foxp2*+ neurons specifically project to the thalamus, Cre recombinase (Cre)-dependent adeno-associated virus (AAV) encoding mGFP and synaptophysin-fused mRuby that labels axon terminals was injected into the mPFC of *Foxp2*-Cre mice (Fig. 1h). The virus expression was confirmed by *post hoc* histological examination (Supplementary Fig. 1f). Consistent with the fact that *Foxp2* labels the L6 CT neurons in the mPFC, whole brain mapping revealed that the mPFC *Foxp2*+ neurons primarily project to thalamus, with the highest intensity of mRuby signals in parataenial thalamic nucleus (PTN), mediodorsal (MD) and ventromedial (VM) thalamus (Fig. 1h). Weaker signals were observed in the nucleus reuniens (Re), paraventricular nucleus of thalamus (PVT) and the medial striatum (Str), while no signal was detected in other downstream targets of mPFC such as NAc, amygdala, PAG, hippocampus, hypothalamus and ventral tegmental area (VTA) (Fig. 1h and Supplementary Fig. 1g–m). Quantification of the average axon-labeling mRuby signals indicated that the medial thalamic nuclei, including PTN, MD, and VM, are the main downstream targets of the mPFC *Foxp2*+ neurons (Fig. 1i). Collectively, these results demonstrate that *Foxp2* specifically labels the mPFC outputs to thalamus (Fig. 1j).

### The mPFC *Foxp2*+ neurons are deactivated in acute and chronic pain

To determine whether the mPFC *Foxp2*+ neurons are involved in pain processing, in vivo miniscopic calcium (Ca2+) imaging was performed to monitor the activity of the individual mPFC *Foxp2*+ neurons under various pain states. To this end, Cre-dependent AAV encoding the Ca2+

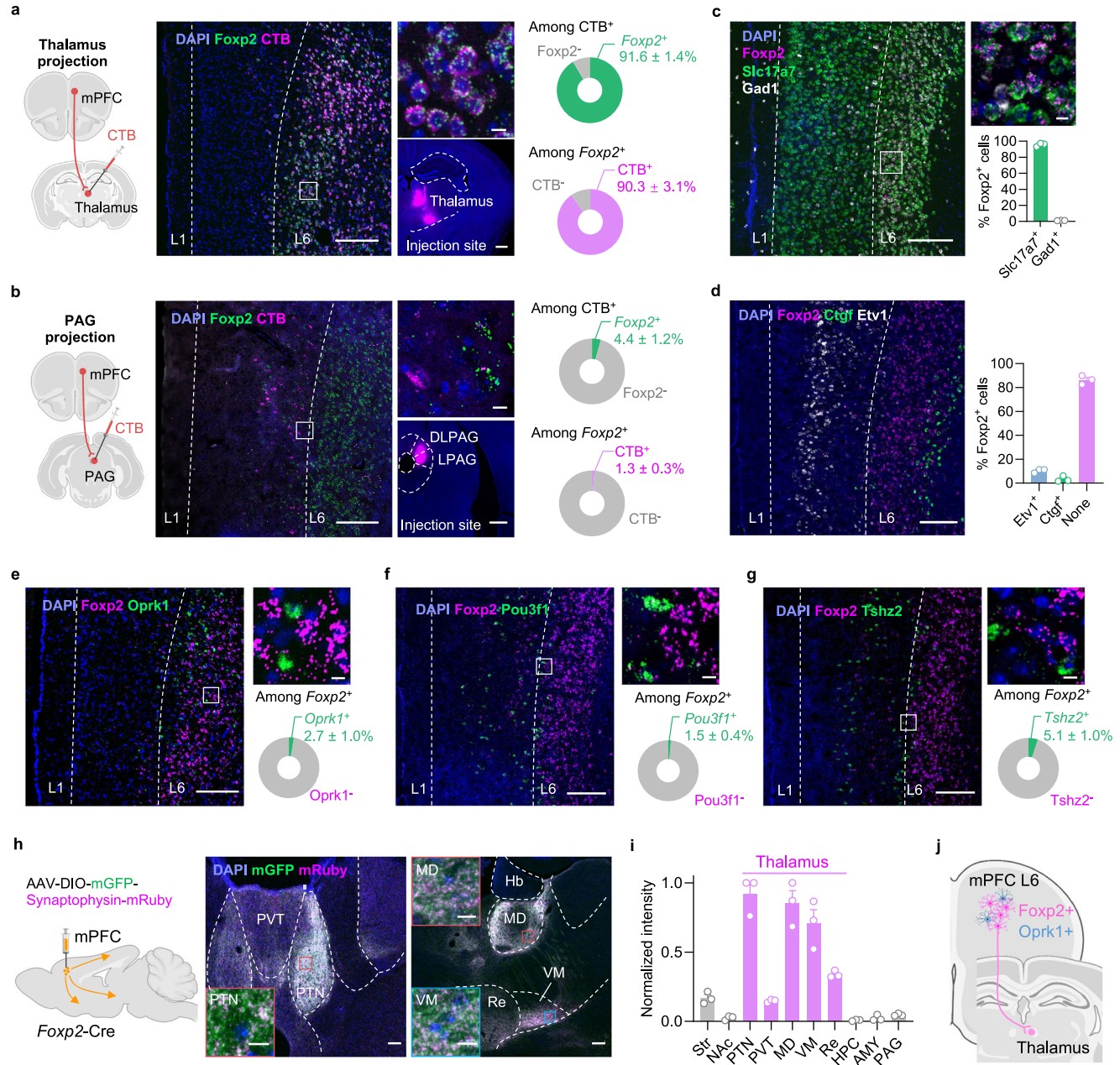

**Fig. 1 | *Foxp2* specifically marks the outputs of mPFC to the thalamus. a** Diagram of CTB injection into the thalamus (left). Middle, representative image of CTB (magenta) and smFISH for *Foxp2* (green) in the mPFC, enlarged image of the indicated region (middle top) and the injection site of CTB (middle bottom). Right, quantification of the CTB⁺ neurons co-expressing *Foxp2* (top right) or the *Foxp2*⁺ neurons co-labeled by CTB (bottom right). Scale bars: middle left, 200 μm; middle top, 10 μm; middle bottom, 500 μm; *n* = 3 mice. **b** Same as (**a**) with CTB injected into the PAG. *n* = 3 mice. **c** Representative image of the smFISH for *Foxp2* (magenta), *Slc17a7* (green), and *Gad1* (white) in the mPFC (left) and enlarged image of the indicated region (top right), as well as the percentages of the *Foxp2*⁺ neurons co-expressing *Slc17a7* or *Gad1* (bottom right). Scale bars: left, 200 μm; top right, 10 μm; *n* = 3 mice. **d** Representative image of smFISH for *Foxp2* (magenta), *Etv1* (white), and *Ctgf* (green) in the mPFC (left) and the percentages of the *Foxp2*⁺ neurons co-expressing *Etv1* or *Ctgf* (right). Scale bar, 200 μm; *n* = 3 mice. Representative images of the smFISH for *Foxp2* (magenta) and *Oprk1* (green) (**e**), *Pou3f1*

(green) (**f**), and *Tshz2* (green) (**g**) in the mPFC (left) and enlarged images of the indicated region (top right), and quantification of the *Foxp2*⁺ neurons co-expressing *Oprk1*, *Pou3f1*, or *Tshz2* in the mPFC (bottom right). Scale bars: left, 200 μm; top right, 10 μm; *n* = 3 mice. **h** Diagram of virus injection (left), and representative images showing the mGFP and mRuby signals in the thalamic subregions (middle, right). Inserted boxes, enlarged images of the indicated regions in PTN, MD, and VM. Scale bars: 100 μm; inserted boxes, 10 μm. **i** Quantification of the mean mRuby fluorescence intensity; *n* = 3 mice. **j** Diagram of the projection pattern of mPFC L6 neurons. Data are represented as mean ± SEM. AMY amygdala, NAc nucleus accumbens, Hb habenula nucleus, HPC hippocampus, MD mediodorsal thalamus, PAG periaqueductal gray, PTN parataenial thalamic nucleus, PVT paraventricular thalamus, Re reuniens nucleus, Str striatum, VM ventromedial thalamus. Source data are provided as a Source Data file. Created in BioRender. Liu, Y. (2026) https://BioRender.com/jpcdwkx.

---

indicator GCaMP7s was injected into the mPFC of *Foxp2*-Cre mice, followed by the implantation of a gradient refractive index (GRIN) lens above the injection site two weeks later (Fig. 2a). Virus expression and proper GRIN lens implantation were confirmed by *post hoc* histological

examination (Fig. 2a). Four weeks after the virus injection, fluorescence signals of the individual *Foxp2*⁺ neurons were recorded through the GRIN lens using a head-mounted miniaturized microscope, and the Ca²⁺ signal traces of individual neurons were identified and extracted

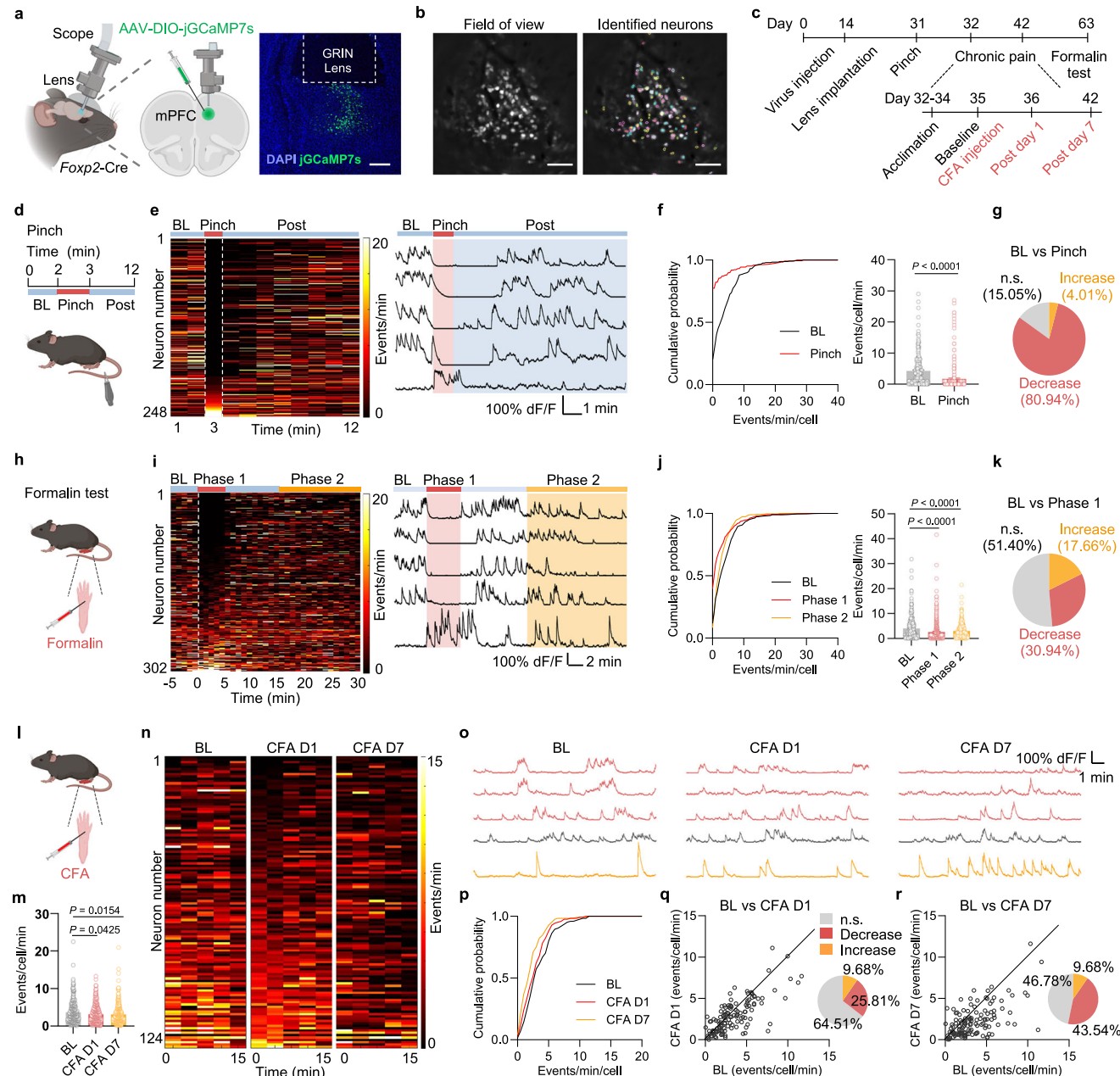

**Fig. 2 | The mPFC *Foxp2*⁺ neurons are deactivated in response to acute and chronic pain. a** Diagram of miniscopic calcium imaging (left) and a representative image of virus expression and lens implantation (right). Scale bar: 200 μm; *n* = 5 mice. **b** An example field of view (left) and identified neuronal contours (right). Scale bars: 100 μm. **c** Timeline of experiments. All the mice were subjected to pinch, CFA, and formalin treatments at the indicated time points. **d** Diagram of pinch stimulation. **e** Left, heatmap of the calcium event frequency averaged to 1 min of bin of all neurons in the pinch experiment (*n* = 248 neurons from 5 mice). The dashed lines indicate the start or end of the pinch, respectively. Right, representative traces of calcium signals (Δ*F/F*) of individual neurons. The red and the blue backgrounds indicate the periods during and after the tail pinch stimulation, respectively. **f** Cumulative distribution (left) and the average (right) of the calcium event frequency in baseline (BL) and pinch stimulation. *n* = 248 neurons from 5 mice. **g** Percentages of neurons with a significant decrease (red), increase (yellow), or no significant (n.s.) change (gray) in the calcium event frequency during pinch stimulation compared to the baseline. **h–k** Same as (**d–g**) but for formalin treatment. *n* = 302 neurons from 5 mice. **l** Diagram of CFA treatment. **m** Averaged

calcium event frequency of all identified neurons in the BL, day 1 and day 7 after CFA injection. *n* = 297, 344, and 335 neurons from 5 mice, respectively. **n** Heatmap of the calcium event frequency averaged to 3 min of bin of the registered same neurons across different days (*n* = 124 neurons from 5 mice). **o** Representative traces of the same neurons with significant decrease (red), increase (yellow), or no significant change (gray) in the frequency in the baseline, day 1, and day 7 after CFA injection. **p** Cumulative distribution of the calcium event frequency of the registered same neurons in baseline, day 1, and day 7 after CFA injection. **q** Comparison of baseline and day 1 events (left); percentages of the neurons with significant decrease (red), increase (yellow), or no significant change (gray) in frequency in day 1 after CFA injection (right). **r** Same as (**q**), except it is for comparison of day 7 after CFA injection. Data are represented as mean ± SEM. Two-tailed paired *t*-test for **f** (*t*₂₄₇ = 5.738), RM one-way ANOVA with Tukey's multiple comparison for **j** ($F_{1.886,567.6}$ = 16.16), and ordinary one-way ANOVA with Tukey's multiple comparison for **m** ($F_{2,973}$ = 4.462). Source data are provided as a Source Data file. Created in BioRender. Liu, Y. (2026) https://BioRender.com/jpcdwkx.

(Fig. 2b). All the mice were subjected to pinch, complete Freund's adjuvant (CFA) and formalin treatments with the Ca²⁺ signals recorded as outline in Fig. 2c.

To examine the activities of the *Foxp2*⁺ neurons in response to noxious mechanical stimulation, we recorded the baseline activity for 2 min, then subjected the mice to tail pinch for 1 min, and monitored an additional 9 min for recovery (Fig. 2d). As shown by the heatmap and representative traces, the Ca²⁺ event frequency of the *Foxp2*⁺ neurons markedly decreased during pinch stimulation, followed by a gradual recovery (Fig. 2e). Quantitative analysis revealed that the overall Ca²⁺ event frequency, but not amplitude, across all *Foxp2*⁺ neurons is significantly decreased during pinch stimulation compared to the baseline (Fig. 2f and Supplementary Fig. 3a), indicating the robust deactivation of the mPFC *Foxp2*⁺ neuronal population. Further analysis of individual neuronal responses showed that around 80% of the *Foxp2*⁺ neurons exhibited a significant decrease in the Ca²⁺ event frequency, with 4% of the neurons showing an increase (Fig. 2g). Together, these results demonstrate that the overall *Foxp2*⁺ neuronal activity is decreased in response to noxious mechanical stimulation.

The mice were also subjected to formalin treatment on the indicated day (Fig. 2c, h), which induces biphasic pain responses as previously described[42]. Enhanced coping behaviors were observed during the first 5 min (phase 1) and between 15 and 30 min (phase 2) following the formalin injection (Supplementary Fig. 3c). Similar to the responses to noxious mechanical stimulation, the Ca²⁺ event frequency of the *Foxp2*⁺ neurons decreased markedly during both phase 1 and phase 2, with a recovery between the two phases (Fig. 2i). Quantitative analysis also revealed that the overall Ca²⁺ event frequency, but not amplitude, across all *Foxp2*⁺ neurons is significantly decreased during phase 1 and phase 2 compared to the baseline (Fig. 2j and Supplementary Fig. 3d, e), suggesting functional deactivation of the mPFC *Foxp2*⁺ neuronal population in response to formalin treatment. Further analysis of individual neuronal responses revealed a significant decrease in Ca²⁺ event frequency in around 30% of *Foxp2*⁺ neurons and increase in around 18% of the neurons (Fig. 2k). To rule out potential effects of prior pain experiences, we repeated the formalin treatment using a new cohort of mice without preceding stimulation (Supplementary Fig. 3f). We observed again a significant decrease in the overall Ca²⁺ event frequency of the *Foxp2*⁺ neurons in both phase 1 and phase 2 (Supplementary Fig. 3g–i), further confirming the deactivation of the mPFC *Foxp2*⁺ neurons in response to formalin treatment. Collectively, these results indicate that the mPFC *Foxp2*⁺ neurons are deactivated in acute nociception.

Finally, we asked whether the mPFC *Foxp2*⁺ neurons are involved in chronic pain. To this end, we subjected the mice to CFA treatment to induce inflammatory pain (Fig. 2l). The overall frequency of Ca²⁺ events in the recorded mPFC *Foxp2*⁺ neurons significantly decreased in both day 1 and day 7 after the CFA injection compared to the baseline (Fig. 2m), indicating a decrease in the *Foxp2*⁺ neuronal activity in CFA-induced inflammatory pain. To better compare the Ca²⁺ events of the individual *Foxp2*⁺ neurons among different days, we identified and analyzed the same neurons recorded in different days (Supplementary Fig. 3f). We found that a large proportion of the *Foxp2*⁺ neurons exhibited decreased Ca²⁺ activities on day 1 and day 7 after CFA injection (Fig. 2n, o). Quantitative analysis revealed the decreased Ca²⁺ events of the registered *Foxp2*⁺ neurons in both day 1 and day 7 (Fig. 2p), and that 25.8% and 43.5% of *Foxp2*⁺ neurons were significantly deactivated on day 1 and day 7, respectively (Fig. 2q, r). Together, these results demonstrate that the mPFC *Foxp2*⁺ neurons are deactivated in CFA-induced inflammatory pain.

## Persistent inactivation of the mPFC *Foxp2*⁺ neurons enhances nociceptive sensitivity, while activation of these neurons alleviates different aspects of pain

Given the critical roles of mPFC and thalamus in pain processing and the observed deactivation of the mPFC *Foxp2*⁺ neurons in both acute and CFA-induced inflammatory pain (Fig. 2), we asked whether these neurons play a causal role in pain regulation. To this end, Cre-dependent AAVs expressing EYFP or tetanus neurotoxin (TeNT) were bilaterally injected into the mPFC of *Foxp2*-Cre mice to persistently block the synaptic output of the *Foxp2*⁺ neurons (Fig. 3a). Behavioral tests were conducted five weeks later, with virus expression confirmed by *post hoc* histological examination (Fig. 3a). We first performed the von Frey test (VFT) to measure the mechanical sensitivity of the mice. The TeNT group exhibited a significantly decreased paw withdrawal threshold compared to the control group, indicating that persistent inactivation of the mPFC *Foxp2*⁺ neurons induces mechanical hypersensitivity of the mice (Fig. 3b). Similarly, a significant decrease in paw withdrawal latency in the hot plate test (HPT) was also observed in the TeNT group, indicating thermal hypersensitivity in these mice (Fig. 3c). Together, these results indicate that persistent inactivation of the mPFC *Foxp2*⁺ neurons enhances nociceptive sensitivity in mice.

We next asked whether short-term manipulation of the mPFC *Foxp2*⁺ neurons affects the nociception. To this end, we bilaterally injected Cre-dependent AAVs encoding mCherry or hM3Dq, an activating DREADD (designer receptors exclusively activated by designer drug), into the mPFC of the *Foxp2*-Cre mice (Fig. 3d). Virus expression was confirmed by *post hoc* histological examination (Fig. 3d). Three weeks later, behavioral assays were performed 20 min after intraperitoneal (i.p.) injection of clozapine-N-oxide (CNO). In both the VFT and HPT, the hM3Dq group showed a significant increase in paw withdrawal threshold and latency (Fig. 3e), indicating the antinociceptive effects induced by the mPFC *Foxp2*⁺ neuron activation. To examine the roles of the mPFC *Foxp2*⁺ neurons in coping behaviors, a formalin test was performed, which revealed a significant decrease in licking duration in phase 2, but not phase 1, in the hM3Dq group (Fig. 3f), further supporting the anti-nociceptive effects of the mPFC *Foxp2*⁺ neuron activation.

We next assessed the roles of the mPFC *Foxp2*⁺ neurons in chronic pain by treating the mice with CFA to induce persistent inflammatory pain. VFT and HPT were performed 2 or 3 days after CFA injection, which revealed significant increases in paw withdrawal threshold and latency in the hM3Dq group (Fig. 3g), indicating activation of the mPFC *Foxp2*⁺ neurons alleviates the mechanical and thermal hypersensitivity in inflammatory pain. To determine the contribution of these neurons to the affective components of pain, we subjected the mice to a conditioned place preference (CPP) test. The mice were conditioned with saline in one chamber and CNO in another (Fig. 3h). After three days of training, the hM3Dq group displayed a preference for the chamber paired with CNO treatment, as indicated by increased time spent in the chamber and a higher CPP score (Fig. 3h), suggesting that activation of the mPFC *Foxp2*⁺ neurons could alleviate affective aspects of inflammatory pain. Collectively, these results indicate that activation of the mPFC *Foxp2*⁺ neurons relieves both nociceptive hypersensitivity and negative affect of inflammatory pain.

To evaluate the effects of short-term inactivation, we bilaterally injected Cre-dependent AAVs expressing mCherry or hM4Di, an inhibitory DREADD, into the mPFC of the *Foxp2*-Cre mice (Supplementary Fig. 4a). Since the overall activity of the mPFC *Foxp2*⁺ neurons has already been decreased under pain conditions, as shown above (Fig. 2 and Supplementary Fig. 3), the chemogenetic inactivation primarily targeted the activated subset of the mPFC *Foxp2*⁺ neurons in response to noxious stimulation. Surprisingly, no significant effects were observed in the nociceptive sensitivity, coping behaviors, or negative affect of pain as indicated by VFT, HPT, formalin test, and CPP test (Supplementary Fig. 4b-d), indicating that the activated subset of *Foxp2*⁺ neurons contributes minimally to pain processing or that its effects are overweighed by the population-wide suppression of *Foxp2*⁺ neurons. The efficiency of chemogenetic manipulation was confirmed as the expression of an immediate-early gene, c-Fos, was decreased or increased after CNO treatment in the hM4Di or hM3Dq group (Supplementary Fig. 4e).

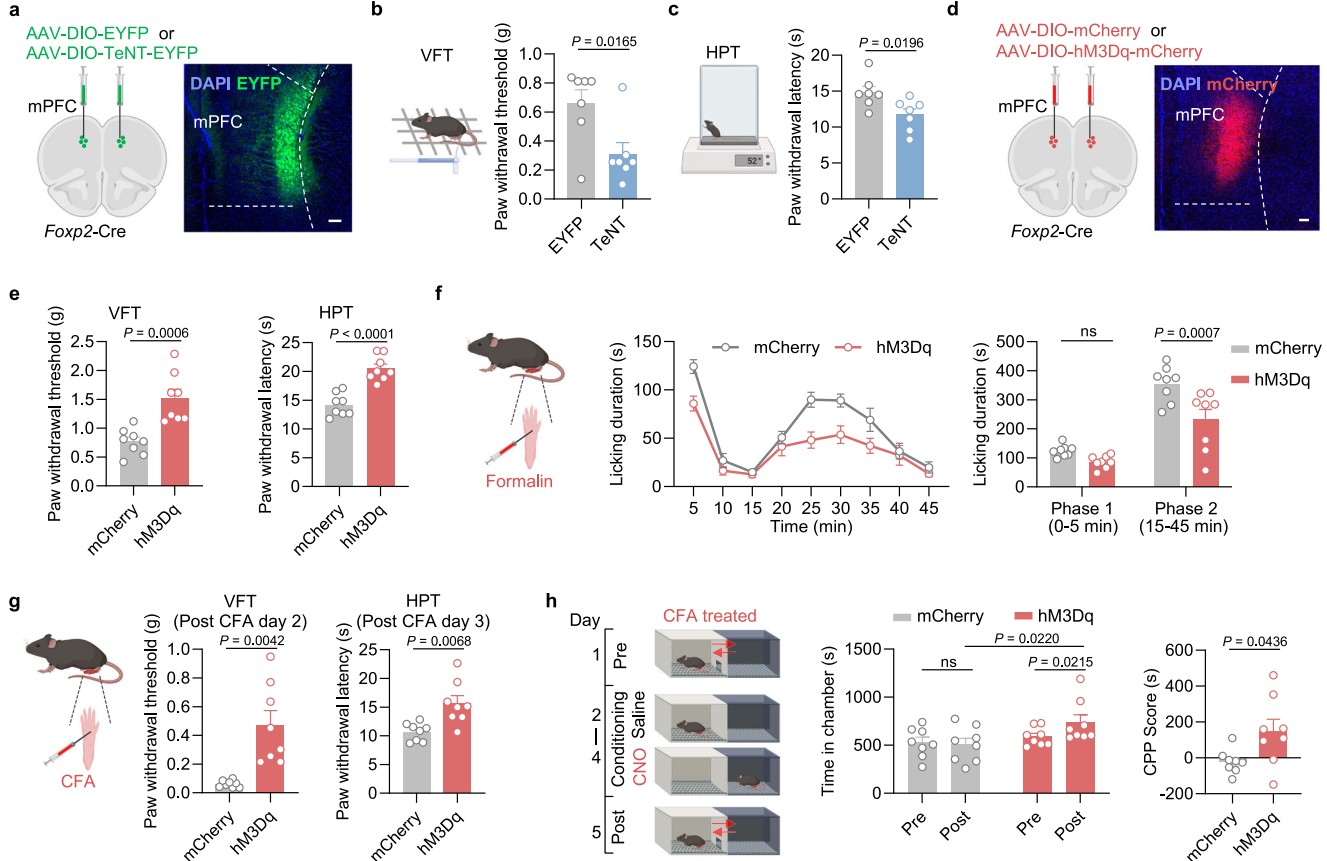

**Fig. 3 | The mPFC *Foxp2*⁺ neurons regulate both sensory and affective aspects of pain. a** Diagram of virus injection (left) and a representative image of TeNT-EYFP expression in the mPFC of *Foxp2*-Cre mice (right). Scale bar: 100 μm. **b** Left, diagram of von Frey test (VFT). Right, paw withdrawal threshold of the mice with EYFP (*n* = 7 mice) or TeNT-EYFP (n = 7 mice) expression assessed in VFT. **c** Diagram of hot plate test (HPT) (left) and paw withdrawal latency of the mice assessed in HPT (right). **d** Diagram of virus injection into the mPFC of *Foxp2*-Cre mice (left) and a representative image of hM3Dq-mCherry expression in the mPFC (right). Scale bar: 100 μm. **e** Paw withdrawal threshold assessed in VFT (left) and latency in HPT (right) of the mice with mCherry or hM3Dq-mCherry expression, respectively. *n* = 8 mice for each group. **f** Diagram of formalin test (left). Licking duration of the mice in response to formalin injection after CNO treatment (middle) and the summarized licking duration in phase 1 and phase 2 (right). **g** Diagram of CFA injection (left) and

paw withdrawal threshold/latency assessed by VFT on post CFA day 2 (middle) and HPT on post CFA day 3 (right). *n* = 8 mice for each group. **h** Diagram of the conditioned place preference (CPP) test starting from post CFA day 5 (left). The time that mice spent in the chamber before and after the conditioning (middle) and CPP scores (right) of the mice in the CPP test. *n* = 8 mice for each group. Data are represented as mean ± SEM. ns no significant difference. Two-tailed unpaired *t*-test for **b** ($t_{12}$ = 2.783), **c** ($t_{12}$ = 2.693) and **e** (VFT, $t_{14}$ = 4.400; HPT, $t_{14}$ = 6.124); RM two-way ANOVA with Sidak's multiple comparison for **f** ($F_{1,14}$ = 12.38) and **h** (middle, $F_{1,14}$ = 3.589); Two-tailed unpaired *t*-test with Welch's correction for **g** (VFT, $t_{7.106}$ = 4.138; HPT, $t_{9.333}$ = 3.459) and **h** (right, $t_{8.663}$ = 2.360). Source data are provided as a Source Data file. Created in BioRender. Liu, Y. (2026) https://BioRender.com/jpcdwkx.

Collectively, these results indicate that persistent inactivation of the mPFC *Foxp2*⁺ neurons enhances nociceptive sensitivity, while activation of these neurons exerts robust antinociceptive effects, alleviating both the sensory and affective components of inflammatory pain.

### Projection-specific roles of the mPFC *Foxp2*⁺ neurons in regulating acute and chronic pain

Having demonstrated the crucial roles of the mPFC *Foxp2*⁺ neurons in regulating pain, we next investigated the underlying circuit mechanisms. We have revealed that the PTN, MD, and VM thalamic subregions are the major downstream targets of the mPFC *Foxp2*⁺ neurons (Fig. 1h, i and Supplementary Fig. 1f–m). Previous studies have revealed the roles of MD and VM in regulating affective, but not sensory aspects of pain[7,22,23], while the PTN is much less studied but implicated in regulating pain sensitivity in mice[43]. To determine the roles of the mPFC *Foxp2*⁺ projections in regulating pain, Cre-dependent AAVs encoding EYFP or channelrhodopsin-2 (ChR2) were injected into the mPFC of *Foxp2*-Cre mice, and optical fibers were implanted above the PTN (Fig. 4a), MD (Fig. 4b), or VM (Fig. 4c), respectively, for projection-specific activation. Virus expression and fiber implantation were

confirmed by *post hoc* histological examination (Fig. 4a–c). Three weeks later, VFT and HPT were conducted. We found that optogenetic activation of the mPFC *Foxp2*⁺ projection to PTN significantly increased the paw withdrawal threshold in VFT and latency in HPT compared to the baseline or the control group (Fig. 4d), indicating reduced mechanical and thermal sensitivity. In contrast, activation of the projections to MD or VM had no significant effect on nociceptive sensitivity (Fig. 4e, f). These results suggest that mPFC *Foxp2*⁺ neurons reduce mechanical and thermal sensitivity through projecting to PTN, but not the MD or VM of the thalamus.

To examine the roles of the projections in coping behaviors under tonic pain, we subjected the mice to the formalin test with optogenetic activation of the specific projections. We found that activation of the projections to PTN and VM, but not MD, significantly reduced the licking duration in phase 2, but not phase 1, of the formalin test compared to the control group (Fig. 4g–i). These results indicate that the projections of the mPFC *Foxp2*⁺ neurons to the PTN and VM reduce the coping behaviors of the mice in response to tonic pain. Collectively, these results demonstrate that the mPFC$^{Foxp2}$ → PTN circuit modulates both the nociceptive sensitivity and coping behavior, and

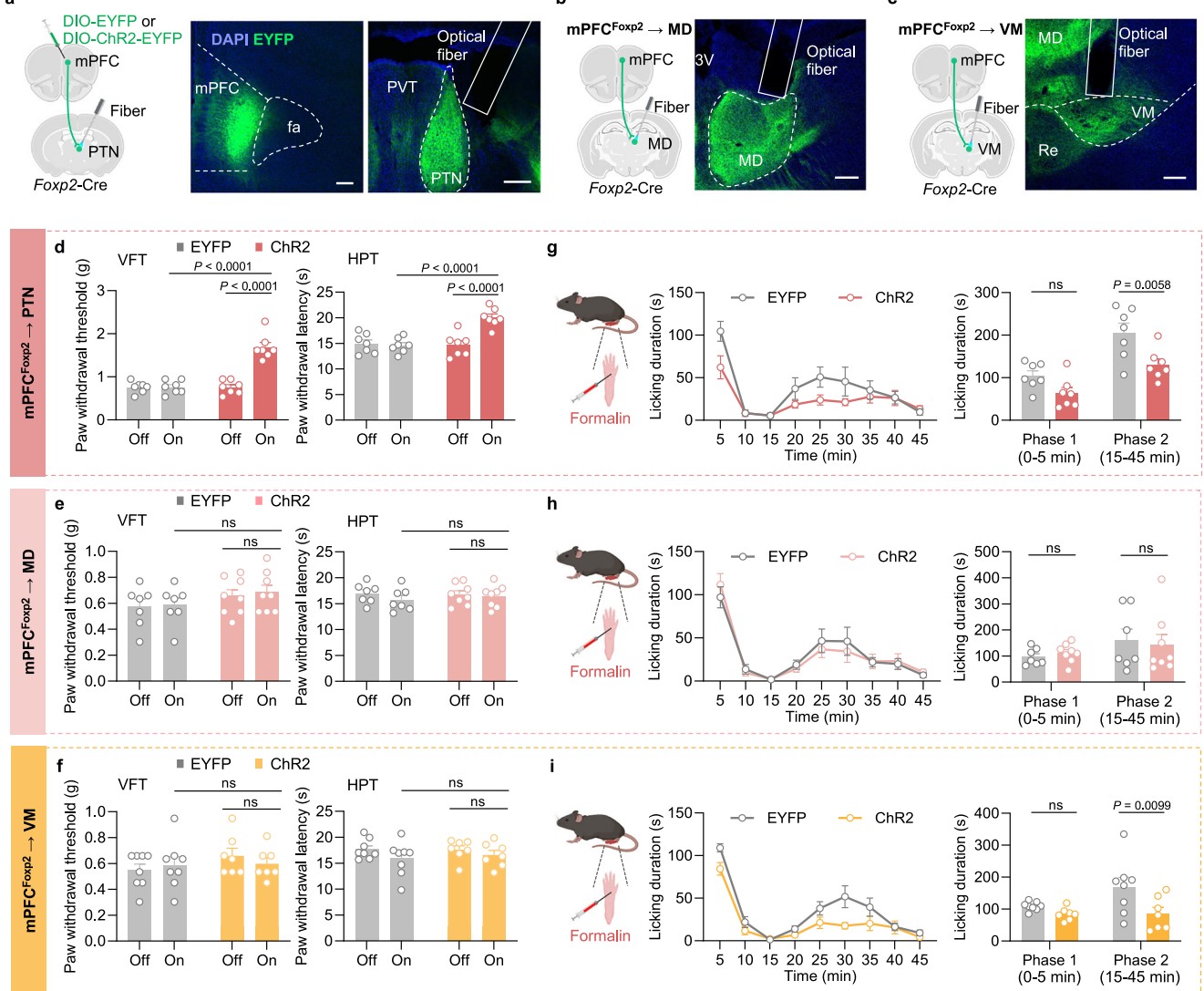

**Fig. 4 | Projection-specific effects of the mPFC *Foxp2*⁺ neurons on acute nociception. a** Diagram of virus injection and optical fiber implantation in the *Foxp2*-Cre mice (left). Representative images showing the injection site in mPFC (middle) and the fiber implantation above the PTN (right). Scale bars: 100 μm. Diagram of surgery in the *Foxp2*-Cre mice (left) and representative images (right) showing the implantation of the optical fiber above the projection to MD (**b**) or VM (**c**). Scale bars: 100 μm. **d** Paw withdrawal threshold/latency assessed by VFT (left) and HPT (right) of the mice expressing GFP (*n* = 7 mice) or ChR2 (*n* = 7 mice) with (on) or without (off) light stimulation targeting the projection to PTN. Same as the (**d**) but targeting the projection to MD (**e**, *n* = 7 mice for EYFP group and 8 mice for ChR2 group) or VM (**f**, *n* = 8

mice for EYFP group and 7 mice for ChR2 group). **g** Diagram of formalin injection (left). Licking duration of the mice in response to formalin injection with light stimulation (middle), and summarized licking duration in phase 1 and phase 2 (right), targeting the projection to PTN (n is the same as in **d**). **h, i** Same as **g** but targeting the projection to MD (**h**, *n* is the same as in **e**) or VM (**i**, *n* is the same as in **f**). Data are represented as mean ± SEM. ns no significant difference. RM two-way ANOVA with Sidak's multiple comparison for (**d**–**i**). $F_{1,12} = 50.45$ (**d**, left), $F_{1,12} = 12.27$ (**d**, right), $F_{1,13} = 1.618$ (**e**, left), $F_{1,13} = 0.1072$ (**e**, right), $F_{1,13} = 0.7602$ (**f**, left), $F_{1,13} = 0.07383$ (**f**, right), $F_{1,12} = 10.64$ (**g**), $F_{1,13} = 0.002235$ (**h**), and $F_{1,12} = 7.024$ (**i**). Source data are provided as a Source Data file. Created in BioRender. Liu, Y. (2026) https://BioRender.com/jpcdwkx.

the mPFC*^Foxp2^* → VM circuit specifically modulates the coping behavior, while the mPFC*^Foxp2^* → MD circuit is not involved in either.

To assess the roles of the projections in chronic pain, we treated the mice with CFA to induce inflammatory pain and performed the pain-related behavioral assays with optogenetic stimulation (Fig. 5a). Two and three days after the CFA injection, VFT and HPT were performed, respectively, to measure mechanical allodynia and thermal hyperalgesia in inflammatory pain. We found that activation of the PTN projection significantly increased the paw withdrawal threshold and latency (Fig. 5b), indicating reduced mechanical and thermal hypersensitivity of the mice. In contrast, no significant change in the VFT and HPT was observed by activating the MD or VM projections (Fig. 5c, d). These results demonstrate that the PTN projection is primarily responsible for the roles of the mPFC *Foxp2*⁺ neurons in alleviating nociceptive hypersensitivity in CFA-induced inflammatory pain.

To investigate the roles of the projections in regulating affective aspects of inflammatory pain, we subjected the mice to the CPP test, pairing one chamber with optogenetic stimulation (Fig. 5a). The mice with activation of the PTN or MD projections exhibited a significant increase in the time spent in the light-paired chamber as well as a higher CPP score (Fig. 5e, f), indicating the alleviated negative affect of pain in these mice. Activation of the VM projection, however, had no significant effects on the chamber preference (Fig. 5g). These results indicate that both the mPFC*^Foxp2^* → PTN and mPFC*^Foxp2^* → MD circuits alleviate the affective aspects of inflammatory pain, while the mPFC*^Foxp2^* → VM circuit is not involved.

Collectively, these results demonstrate that while the mPFC*^Foxp2^* → PTN circuit is involved in regulating nociceptive sensitivity, coping behaviors, and negative affect of pain, the mPFC*^Foxp2^* → MD

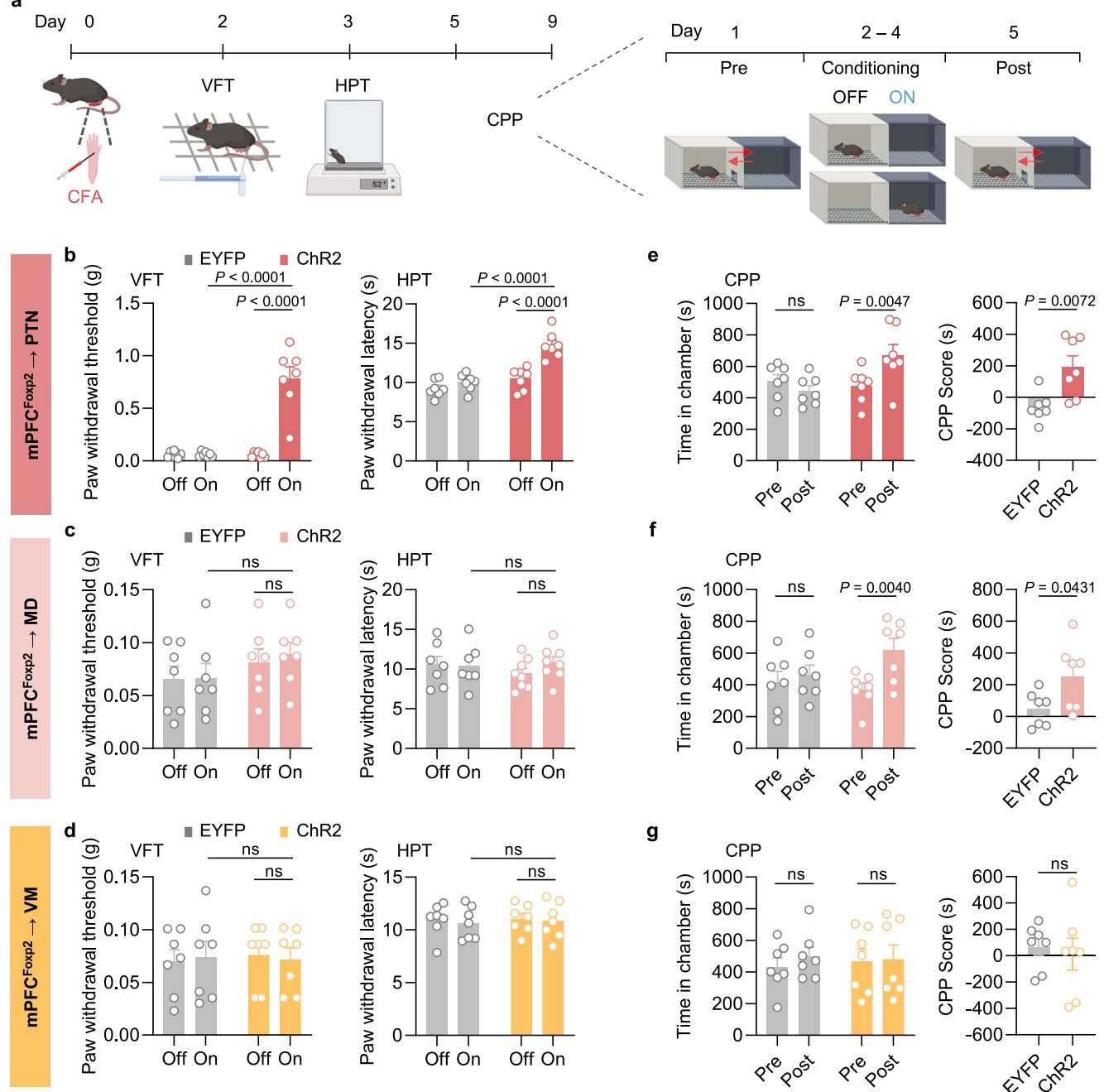

**Fig. 5 | Projection-specific effects of the mPFC *Foxp2*+ neurons on chronic inflammatory pain. a** Schematic diagram of the behavioral tests. **b** Paw withdrawal threshold/latency assessed by VFT (left) and HPT (right) of the mice expressing GFP or ChR2 with (on) or without (off) light stimulation targeting the projection to PTN ($n = 7$ mice for each group). Same as (**b**) but targeting the projection to MD (**c**, $n = 7$ mice for each group) or VM (**d**, $n = 7$ mice for each group). **e** The time spent in the chamber paired with light stimulation (left) and the CPP scores (right) of the mice targeting the projection to PTN. Same as (**e**) but targeting the projection to MD (**f**, $n = 7$ mice for each group) or VM (**g**, $n = 7$ mice for each group). Data are

represented as mean ± SEM. ns, no significant difference. RM two-way ANOVA with Sidak's multiple comparison for **b–d**, **e** (left), **f** (left), and **g** (left); Two-tailed unpaired *t*-test for **e** (right), **f** (right), and **g** (right). $F_{1,12} = 36.01$ (**b**, left), $F_{1,12} = 22.06$ (**b**, right), $F_{1,12} = 1.830$ (**c**, left), $F_{1,12} = 0.06852$ (**c**, right), $F_{1,12} = 0.02375$ (**d**, left), $F_{1,12} = 0.06371$ (**d**, right), $F_{1,12} = 2.968$ (**e**, left), $t_{12} = 3.233$ (**e**, right), $F_{1,12} = 0.5291$ (**f**, left), $t_{12} = 2.261$ (**f**, right), $F_{1,12} = 0.01013$ (**g**, left) and $t_{12} = 0.3941$ (**g**, right). Source data are provided as a Source Data file. Created in BioRender. Liu, Y. (2026) https://BioRender.com/jpcdwkx.

circuit modulates the negative pain affect, and the mPFC*Foxp2* → VM circuit regulates coping behaviors, respectively.

**The mPFC *Foxp2*+ neurons receive antinociceptive cholinergic inputs from HDB**

Having demonstrated the functional deactivation of the mPFC *Foxp2*+ neurons in pain and their pain-relieving effects through projections to distinct thalamic nuclei, we next explored the circuit mechanism

regulating the activity of the mPFC *Foxp2*+ neurons. Previous studies have shown that mPFC receives intense cholinergic innervation from the basal forebrain and is strongly modulated by acetylcholine (ACh)[26,44–46]. Notably, cholinergic modulation of the mPFC is severely impaired in neuropathic pain, while activating the cholinergic projection from the nucleus basalis magnocellularis to the mPFC induces antinociceptive effects[47,48]. These observations prompted us to investigate whether the mPFC *Foxp2*+ neurons are modulated by cholinergic innervation.

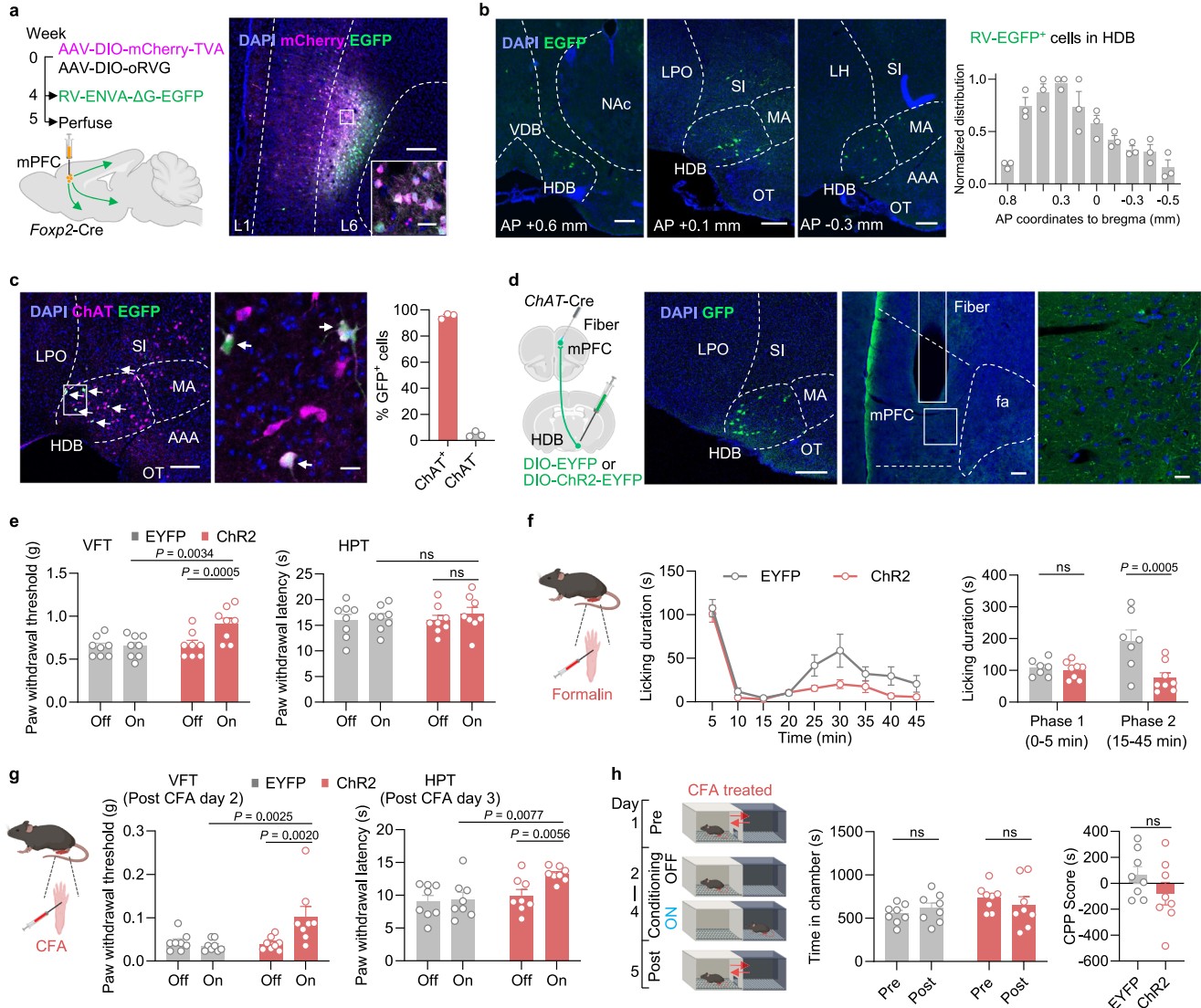

**Fig. 6 | The mPFC *Foxp2*⁺ neurons receive cholinergic innervation from HDB, which is antinociceptive. a** Diagram of virus injection for rabies tracing (left). Representative image of virus expression in the injection site and magnified image of the indicated region (right). Scale bars: 200 μm; inserted box, 20 μm. *n* = 4 mice. **b** Representative images (left) and quantification (right) of the RV-EGFP labeled cells from the anterior to posterior HDB. Scale bars: 200 μm. *n* = 3 mice. **c** Immunostaining of ChAT (left) and enlarged image of the indicated region (middle). Quantification of the GFP⁺ cells co-expressing ChAT in HDB (right). Scale bars: left, 200 μm; middle, 20 μm. *n* = 3 mice. **d** Diagram of the experiment (left), representative images of virus expression in HDB and fiber implantation in mPFC (middle), and magnified image of the indicated region (right). Scale bars, middle, 200 μm; right, 20 μm. **e** Paw withdrawal threshold assessed in VFT (left) and latency in HPT (right) of the mice expressing EYFP or ChR2 with (on) or without (off) light stimulation. *n* = 8 mice for each group. **f** Diagram of formalin injection (left). Licking

duration of the mice with light stimulation (middle), and the summarized licking duration in phase 1 and phase 2 (right). *n* = 7 mice for the EYFP group and 8 mice for the ChR2 group. **g** Diagram of CFA injection (left), and the paw withdrawal threshold (middle) and latency (right) of the mice with or without light stimulation. *n* = 8 mice for each group. **h** Diagram of CPP test (left). The time that the mice spent in the chamber before and after paired with light stimulation (middle), and CPP scores of the mice (right). *n* = 8 mice for each group. Data are represented as mean ± SEM. ns no significant difference. RM two-way ANOVA with Sidak's multiple comparison for **e–g**, and **h** (left); Two-tailed unpaired *t*-test for **h** (right). $F_{1,14} = 4.532$ (**e**, left), $F_{1,14} = 0.09970$ (**e**, right), $F_{1,13} = 8.399$ (**f**, left), $F_{1,14} = 4.275$ (**g**, left), $F_{1,14} = 5.240$ (**g**, right), $F_{1,14} = 2.108$ (**h**, left), and $t_{14} = 1.336$ (**h**, right). Source data are provided as a Source Data file. Created in BioRender. Liu, Y. (2026) https://BioRender.com/jpcdwkx.

To determine whether the mPFC *Foxp2*⁺ neurons receive direct monosynaptic cholinergic inputs, rabies virus (RV)-based monosynaptic retrograde tracing was performed (Fig. 6a). Cre-dependent AAV helpers encoding TVA-mCherry and rabies G protein were injected into the mPFC of *Foxp2*-Cre mice, followed by EnvA-ΔG rabies virus expressing EGFP injected into the same region 4 weeks later. One week after the RV injection, the virus expression and starter cells were confirmed in the mPFC (Fig. 6a). Brain-wide analysis revealed RV-EGFP-labeled neurons in several brain regions, including the basal forebrain, insular cortex, thalamus, ventral hippocampus, and basolateral amygdala, while fewer labeled neurons were also observed in the

hypothalamus, dorsal raphe nucleus, and parabrachial nucleus (Fig. 6b and Supplementary Fig. 5a, b). Notably, a substantial number of neurons were labeled with RV-EGFP in the basal forebrain, particularly the HDB, with the RV-EGFP-labeled neurons distributed from the anterior to the posterior part of HDB (Fig. 6b). Further immunostaining showed that most RV-EGFP-labeled neurons in HDB also express choline acetyltransferase (ChAT), a marker for cholinergic neurons (Fig. 6c), indicating the direct monosynaptic cholinergic inputs from HDB to the mPFC *Foxp2*⁺ neurons. These results suggest that the mPFC *Foxp2*⁺ neurons receive monosynaptic inputs from the HDB cholinergic neurons.

To examine the functions of the HDB inputs to the mPFC in pain, Cre-dependent AAVs encoding EYFP or ChR2 were injected into the HDB of the *ChAT*-Cre mice and an optical fiber was implanted above the mPFC region to enable specific activation of the HDB$^{ChAT}$ → mPFC circuit (Fig. 6d). The virus expression as well as the fiber implantation were confirmed by *post hoc* histological examination (Fig. 6d). Three weeks after the surgery, VFT and HPT were performed and a significant increase in the VFT was observed in response to optogenetic activation of the circuit (Fig. 6e), indicating reduced mechanical sensitivity of the mice. However, no significant effect was observed in the HPT (Fig. 6e), indicating that the baseline thermal sensitivity is unaffected. To assess the role of this circuit in coping behaviors under tonic pain, the mice were subjected to formalin test (Fig. 6f). We found that the ChR2 group of mice exhibited significantly reduced licking duration in phase 2, but not phase 1, after formalin treatment (Fig. 6f), indicating that activation of the HDB cholinergic inputs to mPFC reduces the coping behaviors in mice. To further evaluate the roles of this circuit in inflammatory pain, the mice were subjected to CFA treatment. VFT and HPT analyses indicated that optogenetic activation of the HDB cholinergic terminals in mPFC significantly alleviated both the mechanical and thermal hypersensitivity in inflammatory pain, as indicated by increased paw withdrawal threshold and latency (Fig. 6g). We next performed the CPP test to measure the negative effect of pain, and surprisingly, no significant effect was observed (Fig. 6h), indicating that activation of the HDB cholinergic inputs to mPFC does not alleviate the affective components of inflammatory pain. Together, these results demonstrate that the mPFC *Foxp2*$^+$ neurons receive direct monosynaptic cholinergic innervation from the HDB, which regulates the nociceptive sensitivity and coping behaviors.

## The mPFC *Foxp2*$^+$ neurons highly express α4β2 nAChR, which facilitates pain relief

Given the critical roles of the cholinergic signaling in pain processing and the innervation of mPFC *Foxp2*$^+$ neurons by cholinergic inputs, we next sought to identify the specific acetylcholine receptor (AChR) responsible for modulating the mPFC *Foxp2*$^+$ neurons. In the cerebral cortex, there are mainly two types of nicotinic acetylcholine receptors (nAChR), the α7 (encoded by *Chrna7*) and the α4β2 (encoded by *Chrna4* and *Chrnb2*) nAChRs, and two types of muscarinic acetylcholine receptors (mAChR), the M1 (encoded by *Chrm1*) and M2 receptors (encoded by *Chrm2*)[46]. To this end, we performed smFISH analyses for these receptors as well as *Foxp2* and *Oprk1*. The results showed that over 95% of the *Foxp2*$^+$ cells express *Chrna4* while very few express *Chrna7* (Fig. 7a). Further smFISH analysis of *Chrna4* and *Chrnb2* revealed that around 93% of the *Foxp2*$^+$ cells are positive for both (Supplementary Fig. 6a), while less than 10% of *Oprk1*$^+$ cells express both (Fig. 7b and Supplementary Fig. 6b), indicating the predominant expression of α4β2 nAChR in the *Foxp2*$^+$ neurons. smFISH analysis of mAChRs showed that *Chrm1* was widely expressed across the mPFC, with expression in nearly all *Foxp2*$^+$ and *Oprk1*$^+$ cells in L6 (Fig. 7c and Supplementary Fig. 6c), while minimal expression of *Chrm2* in *Foxp2*$^+$ cells were observed (Fig. 7d and Supplementary Fig. 6d). Notably, quantitative analysis revealed the selective expression of the α4β2 nAChRs in *Foxp2*$^+$ cells in mPFC L6, with nearly 80% of α4β2 nAChR-expressing cells being *Foxp2*$^+$ and only a minority is *Oprk1*$^+$ (Fig. 7e). Collectively, these results indicate that while α7 nAChR and M2 expressions are relatively low and M1 is broadly expressed in mPFC, the α4β2 nAChR is highly expressed and enriched in the L6 *Foxp2*$^+$ neurons of mPFC (Fig. 7f).

Given the critical roles of the mPFC *Foxp2*$^+$ neurons in pain processing and the strong effects of ACh in modulating neuronal activity in mPFC[46], we next asked whether the α4β2 nAChR in mPFC regulates pain. To this end, we bilaterally implanted a guided cannula for intracranial drug infusion into the mPFC of WT mice, confirmed by *post hoc* histological examination (Fig. 7g). We first infused artificial

cerebrospinal fluid (ACSF) or ABT-594, a specific α4β2 nAChR angonist[27], at different dosages (2.5, 5, and 25 pmol/site) into the mPFC and performed VFT to measure the mechanical sensitivity of the mice. We found that ABT-594 at 5 or 25 pmol per site significantly increased the paw withdrawal threshold in VFT compared to both baseline and the ACSF control (Fig. 7h and Supplementary Fig. 7a), indicating reduced mechanical sensitivity by activating the α4β2 nAChR. smFISH of *Foxp2* and *cfos* after ACSF or ABT-594 (5 pmol per site) treatment also revealed a significant increase in the *cfos* expression in the mPFC *Foxp2*$^+$ neurons (Supplementary Fig. 7b), indicating that ABT-594 at this dosage is sufficient to activate the mPFC *Foxp2*$^+$ neurons.

Since both the antinociceptive effect and neuronal activation are sufficiently elicited by the treatment of ABT-594 at the dosage of 5 pmol per site, this dosage is used in the subsequent experiments. Similar antinociceptive effects were observed in the hot plate test (HPT), with increased paw withdrawal latency following ABT-594 treatment (Fig. 7h). To test whether the α4β2 nAChR is also involved in regulating chronic pain, we treated the mice with CFA to induce inflammatory pain. We found that ABT-594 treatment significantly increased the paw withdrawal threshold in VFT and latency in HPT (Fig. 7i), indicating the treatment alleviated mechanical and thermal hypersensitivity in inflammatory pain. However, CPP tests revealed no significant effect of ABT-594 treatments (5 or 25 pmol per site) on the affective components of pain (Supplementary Fig. 7c–e), indicating that ABT-594 treatment is not sufficient to relieve the affective components of pain.

The high levels of α4β2 nAChR expression in the mPFC *Foxp2*$^+$ neurons prompted us to ask whether the antinociceptive effects of ABT-594 depend on activation of the mPFC *Foxp2*$^+$ neurons. To this end, Cre-dependent AAVs encoding mCherry or hM4Di were bilaterally injected into the mPFC of the *Foxp2*-Cre mice, followed by guided cannula implantation for intracranial drug delivery (Fig. 7j). The virus expression and the cannula implantation were confirmed by *post hoc* histological examination (Fig. 7j). Three weeks after the surgery, VFT and HPT were performed. In both the mCherry and hM4Di groups, intracranial ABT-594 delivery significantly increased the paw withdrawal threshold in VFT and latency in HPT (Fig. 7k). Notably, systemic administration of CNO, which selectively inhibits hM4Di-expressing mPFC *Foxp2*$^+$ neurons, abolished these effects in the hM4Di group, while no significant changes were observed in the mCherry group (Fig. 7k). These results indicate that the antinociceptive effects of α4β2 nAChR activation are mediated by the mPFC *Foxp2*$^+$ neuron activity.

Collectively, these results demonstrate that the α4β2 nAChR is highly expressed in the mPFC *Foxp2*$^+$ neurons and activation of this receptor is antinociceptive, which depends on the activation of the mPFC *Foxp2*$^+$ neurons.

## Discussion

The present study reveals how the mPFC modulates different dimensions of pain through different thalamic projections and uncovers the critical roles of the cholinergic signaling in mPFC in pain regulation (Fig. 8). We identified *Foxp2* as a marker for the thalamus-projecting neurons in mPFC by neural tracing and smFISH (Fig. 1). In vivo recordings showed that these neurons are deactivated in both acute and chronic pain (Fig. 2). Combined cell-type-specific and circuit-specific manipulation studies revealed the critical roles of the mPFC *Foxp2*$^+$ neurons in regulating sensory and affective pain through discrete thalamic projections (Figs. 3–5). We also showed the cholinergic innervation from HDB and the expression of α4β2 nAChR, which is antinociceptive, in the mPFC *Foxp2*$^+$ neurons, underscoring the roles of the HDB$^{ChAT}$→mPFC$^{Foxp2}$→thalamus circuit in pain regulation (Figs. 6 and 7).

The cerebral cortex is organized into layers with an intermingled distribution of projection neurons that target diverse brain regions[37]. While our understanding of cortical circuitry has predominantly emerged from sensory cortices[19,49], recent studies have revealed some

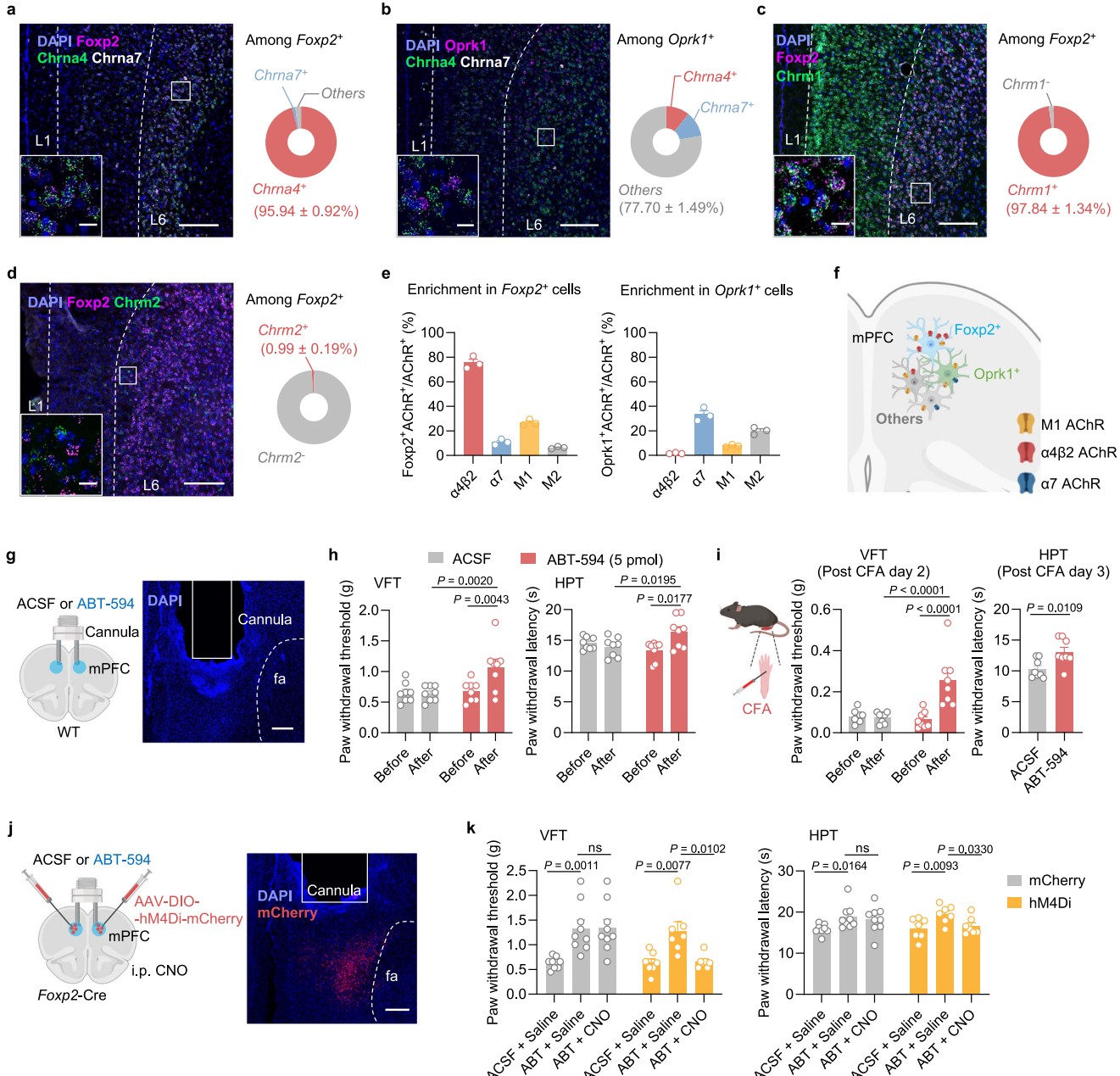

**Fig. 7 | Cell-type-specific expression pattern of acetylcholine receptors in mPFC, and the α4β2 nAChR relieves pain through the *Foxp2*⁺ neurons in mPFC. a** Left: smFISH for *Foxp2* (magenta), *Chrna4* (green), and *Chrna7* (white) in the mPFC and the enlarged image of the indicated region. Right: quantification of *Foxp2*⁺ cells co-expressing *Chrna4* or *Chrna7*. Scale bars: 200 μm; inserted box, 20 μm. *n* = 3 mice. Same as (**a**) but for *Oprk1* (magenta), *Chrna4* (green), and *Chrna7* (white) (**b**), for *Foxp2* (magenta) and *Chrm1* (green) (**c**) or for *Foxp2* (magenta) and *Chrm2* (green) (**d**). *n* = 3 mice. **e** Percentages of the *Foxp2*⁺ cells (left) and the *Oprk1*⁺ cells (right) among the cells expressing different types of acetylcholine receptor. *n* = 3 mice. **f** Summary of acetylcholine receptor expressions in the mPFC. **g** Diagram (left) and validation (right) of the cannula implantation into the mPFC. Scale bar: 200 μm. **h** Paw withdrawal threshold in VFT (left) and latency in HPT (right) of the mice before and after injection of ACSF or ABT-594 (5 pmol/site) into

the mPFC. *n* = 8 mice for each group. **i** Diagram of CFA injection (left), paw withdrawal threshold in VFT (middle), and latency in HPT (right) of the mice in response to intracranial injection of ABT−594 (animal number is the same as in **h**). **j** Diagram (left) and validation (right) of the virus injection and cannula implantation. Scale bar: 200 μm. **k** Paw withdrawal threshold in VFT (left) and latency in HPT (right) of the mice in response to ABT-594 treatment and CNO treatment. *n* = 9 mice for the mCherry group and 7 mice for the hM4Di group. Data are represented as mean ± SEM. ns, no significant difference. RM two-way ANOVA with Sidak's multiple comparison for **h**, **i** (left), and **k**; Two-tailed unpaired *t*-test for **i** (right). $F_{1,14} = 6.217$ (**h**, left), $F_{1,14} = 1.419$ (**h**, right), $F_{1,14} = 8.431$ (**i**, left), $t_{14} = 2.934$ (**i**, right), $F_{2,28} = 13.39$ (**k**, left), $F_{2,28} = 9.794$ (**k**, right). Source data are provided as a Source Data file. Created in BioRender. Liu, Y. (2026) https://BioRender.com/jpcdwkx.

distinct features of cellular organization and connections of the mPFC[19,35,36,50]. Notably, we found that *Ntsr1*, a widely used marker for L6 CT neurons in sensory cortex and motor cortex[22,33,34], is broadly expressed across multiple layers in mPFC. The expression of *Foxp2* has been well described in the developing brain and the sensory cortex of the adult brain[37–40], but it is not clear whether *Foxp2* is specifically

expressed in the L6 CT neurons in the mPFC of adult mice. Our smFISH results showed that the *Foxp2* expression is restricted in the glutamatergic neurons of the deep layers in mPFC, mostly in L6, and has few overlaps with the markers for other projection neurons such as IT, ET, and NP neurons (Fig. 1c–g). Retrograde labeling further confirmed that *Foxp2* specifically labels the mPFC neurons projecting to the thalamus,

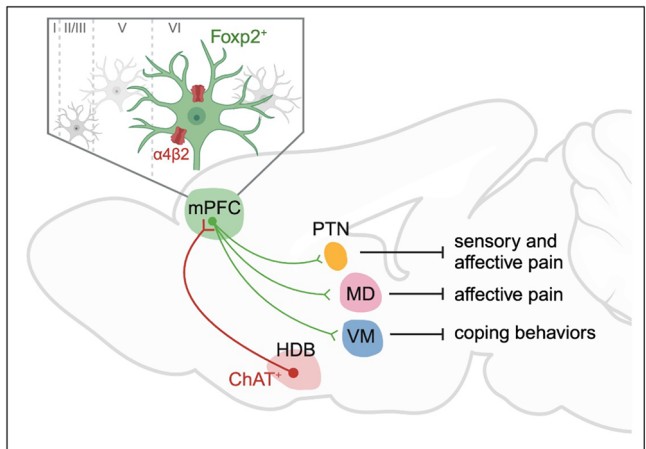

**Fig. 8 | Graphic summary of the HDB^ChAT→mPFC^Foxp2→thalamus connections in regulating different aspects of pain.** In the mPFC, the major outputs to the thalamus originate from the corticothalamic neurons, which are specifically marked by the expression of *Foxp2*. These neurons are drastically deactivated in response to various acute noxious stimuli and in chronic inflammatory pain, and activation of these neurons regulates different aspects of pain through segregated thalamic projections. The mPFC *Foxp2*+ neurons that project to PTN are involved in relieving both the sensory and affective aspects of pain, while the projection to MD relieves affective pain and the one to VM regulates coping behaviors, respectively. The mPFC *Foxp2*+ neurons receive cholinergic innervation from the HDB and highly express the α4β2 nAChR, which relieves pain by activating the *Foxp2*+ neurons in mPFC. Created in BioRender. Liu, Y. (2026) https://BioRender.com/jpcdwkx.

but not those to cPFC, PAG, NAc, or BLA (Fig. 1a, b and Supplementary Fig. 1c–e). Consistently, axon terminal labeling revealed that the mPFC *Foxp2*+ neurons almost exclusively project to the thalamus with much weaker signals in the medial striatum, and no detectable signal in NAc, BLA, and PAG (Fig. 1h, i and Supplementary Fig. 1f–m). Interestingly, it is reported that a subset of *Foxp2*+ neurons in PFC project to VTA[39], but we did not observe axon-labeling signals in VTA (Supplementary Fig. 1 m). One possible explanation is that the reported VTA-projecting *Foxp2*+ neurons are mainly located in the ventral part of PFC, especially the dorsal peduncular nucleus (DPn) which show different projection patterns from that of the mPFC[51]. Moreover, we observed a limited number of neurons in L5 that express *Foxp2* and found that these neurons do not express markers of IT, ET, and NP neurons in mPFC (Fig. 1e, g), consistent with tracing results. Our study, therefore, provides multiple lines of evidence supporting *Foxp2* as a specific marker for the thalamus-projecting neurons in mPFC.

The mPFC *Foxp2*+ neurons are predominantly L6 CT neurons and constitute the major output of mPFC to thalamus (Fig. 1 and Supplementary Fig. 1). In contrast, the L6 IT neurons do not project to the thalamus but instead target intratelencephalic regions such as the contralateral mPFC and NAc. It has been reported that the L5 ET neurons also project to the thalamus, albeit to a lesser extent[52]. Both the L5 ET and L6 CT neurons project to the MD and VM, with the L6 CT neurons acting as modulators that evoke robust firing of thalamo-cortical neurons through facilitating synapses[53,54]. Outputs of the mPFC to other brain regions, such as NAc and amygdala, arise primarily from the IT neurons[19]. L5 ET projection to the PAG has been well described in pain regulation through the descending modulatory pathway[16,17], whereas IT neurons projecting to the NAc and amygdala modulate the affective dimensions of pain and associated comorbidities[55,56]. Our study reveals a critical role of the mPFC L6 CT neurons in regulating both sensory and affective aspects of pain via discrete thalamic projections, thus filling in an important gap in understanding the contribution of the mPFC in pain processing.

Substantial evidence supports the involvement and critical roles of mPFC in pain regulation[8,10], and previous studies have uncovered

the roles of the mPFC L5 ET neurons projecting to PAG, a key node for the descending modulatory pathway that dictates the processing of noxious information in the spinal cord[16,17]. However, although the functions of the mPFC IT neurons that project to BLA and striatum have also been indicated[8,9], it is much less clear how the mPFC modulates the processing of noxious information in the brain, particularly for the roles of the output to thalamus[30]. The thalamus is a key hub for integrating noxious information and regulates multiple aspects of pain[5]. We showed that the thalamus-projecting *Foxp2*-expressing neurons in mPFC are deactivated in response to various noxious stimuli and recover gradually after the removal of the stimuli (Fig. 2), indicating the involvement of these neurons in both acute and chronic pain. In contrast to the motor cortex to thalamus circuit that is mainly involved in affective pain[22], we found that persistent inactivation of the mPFC *Foxp2*+ neurons sensitizes nociception (Fig. 3a–c), while activation relieves both sensory and affective components of pain (Fig. 3d–h), demonstrating the crucial functions of the mPFC CT neurons in the endogenous pain relief pathway.

Our study also delineates the circuit mechanism underlying the roles of the mPFC *Foxp2*+ neurons in regulating pain. The thalamus receives nociceptive information directly from the spinothalamic tract and is widely connected to other brain regions[57]. It has been proposed that the lateral thalamic system mediates discriminative features of pain, while the medial system mediates the emotional aspects of pain[5]. Our results show PTN, MD, and VM of the thalamus as the major targets of the mPFC *Foxp2*+ neurons (Fig. 1h, i) and reveal the critical roles of the mPFC in modulating the nociceptive processing by these thalamic projections (Figs. 4 and 5). We found that while the projection of the mPFC *Foxp2*+ neurons to PTN regulates both sensory and affective pain, the projection to MD relieves affective pain, and the one to VM regulates coping behaviors specifically (Figs. 4 and 5). These findings support the idea that segregated circuit mechanisms are responsible for different dimensions of pain[5,20]. Our results reveal PTN, which is a relatively less studied region with the functions and circuits remaining to be fully characterized, as a master hub in the medial thalamus for regulating both the sensory and affective aspects of pain. Future study is needed to further clarify the circuit mechanism underlying the critical roles of PTN in pain regulation. In summary, our study reveals the critical roles and the segregated circuit mechanisms of mPFC in governing the nociceptive processing in the thalamus.

The mPFC is broadly connected to a wide range of brain regions[19]. While the strengthened input from BLA to the inhibitory neurons in mPFC has been proposed to be a key mechanism underlying the functional deactivation of mPFC in chronic pain[17], the contributions of other inputs are far from fully understood. The mPFC receives dense cholinergic innervation from the basal forebrain and expresses high levels of cholinergic receptors[26,46]. It is reported that the ACh receptors are expressed in a layer-specific manner, with some expression of α4β2 nAChR in the inhibitory neurons of L2/3 and much higher expression in the excitatory neurons in L5/6 of mPFC[44]. Consistently, our results showed that the expression of α4β2 nAChR is enriched in deep layers of mPFC and more specifically in the L6 CT neurons, while the α7 nAChR is expressed in IT neurons (Fig. 7a–f and Supplementary Fig. 6). We further revealed that the cholinergic inputs to the *Foxp2*+ neurons are from the basal forebrain, particularly the HDB, by monosynaptic retrograde tracing (Fig. 6a–c and Supplementary Fig. 5). Our study thus revealed the cell-type-specific expression patterns of the AChRs in mPFC as well as the cholinergic inputs to the mPFC L6 CT neurons.

Cholinergic signaling has been extensively studied in the context of pain and analgesia, with significant insights into its roles in the peripheral, spinal cord, and descending modulation pathway[58–61]. However, the functions and circuit mechanisms of the cholinergic system in the brain in nociceptive processing are not fully understood[47]. While cholinergic input is known to be essential for the mPFC functions such as attention and cognition, its role in pain

regulation is less clear[26,45,46]. Notably, cholinergic signaling in the mPFC is markedly impaired in neuropathic pain models[48], and activation of cholinergic projections from the nucleus basalis magnocellularis to the mPFC has been shown to produce antinociceptive effects[47]. Our study revealed that the mPFC $Foxp2^+$ neurons receive cholinergic inputs from the HDB, which regulates both sensory pain and coping behaviors (Fig. 6d–h). We also found that targeting the α4β2 nAChR in mPFC with a specific agonist ABT-594, which is antinociceptive through systemic administration or brainstem injection[27], alleviates pain (Fig. 7g–i). Importantly, this antinociceptive effect requires the activity of the mPFC $Foxp2^+$ neurons (Fig. 7j, k). Notably, desensitization is an important property of the nAChRs, and the receptors are desensitized in response to high concentration or affinity of agonist[62]. In our study, we used a relatively low concentration of agonist that has been found to activate the neurons in the brainstem[27]. On the other hand, the L6 pyramidal neurons in the cortex express $Chrna5$, which encodes an accessory subunit of the α5-containing α4β2 nAChR[63]. This receptor complex could increase conductance and prolong inward currents in response to persistent nicotine application and is important for the tonic effects of ACh[46,64]. Thus, the α4β2 nAChR in $Foxp2^+$ neurons could play an important role in maintaining the baseline neuronal activity and activating the endogenous pain relief pathway. While further studies are still needed to fully elucidate how the cholinergic inputs from HDB modulate the mPFC activity in chronic pain, our results support the crucial role of the cholinergic signaling in the mPFC $Foxp2^+$ neurons in pain processing.

Besides the cholinergic inputs, we also revealed other monosynaptic inputs to the mPFC L6 CT neurons, including the IC and BLA (Supplementary Fig. 5), which are important for pain processing[5,65]. It is well-known that in chronic pain, BLA sends strengthened feedforward inhibition to the L5 ET neurons of mPFC that further project to the PAG to regulate pain[17]. However, the effects of the BLA inputs to the L6 CT neurons are unknown. Future studies will reveal the respective contributions of these specific inputs.

Collectively, our study uncovers the crucial roles of the mPFC in modulating sensory and affective aspects of pain through the discrete $HDB^{ChAT}$→$mPFC^{Foxp2}$→thalamic nuclei circuits and raises the possibility of managing chronic pain by targeting the mPFC L6 CT neurons.

## Methods
### Animals
All experiments were conducted in accordance with the National Institutes of Health Guide for Care and Use of Laboratory Animals and approved by the Institutional Animal Care and Use Committee (IACUC) of Boston Children's Hospital and Harvard Medical School. The wild-type C57BL/6 mice (Jax, 000664), B6.Cg-$Foxp2^{tm1.1(cre)Rpa}$/J mice (Jax, 030541) and B6.129S6-$Chat^{tm2(cre)lowl}$/J mice (Jax, 006410) were purchased from Jackson Laboratory. Mice were housed with littermates (3–5 mice/cage) in a 12-h light/dark cycle (light on from 7:00 to 19:00) with food and water $ad$ $libitum$ unless otherwise specified. Ambient temperature (23–25 °C) and humidity (55–62%) were automatically controlled. For the present study, 2–4-month-old male mice were used, and all mice were randomly assigned to different groups. All the independent behavioral assays were performed on separate days to minimize stress and avoid potential confounding effects between experiments.

### Single-molecule fluorescence in situ hybridization (smFISH) and immunofluorescence (IF) staining
Mice were euthanized by inhalation of $CO_2$ and transcardially perfused with PBS and 4% paraformaldehyde (PFA) before the brains were collected and fixed in 4% PFA at 4 °C overnight, followed by dehydration in 30% sucrose solution for 3 days. The brains were then frozen in Optimal Cutting Temperature embedding media (Sakura, #4583), and coronal sections (14 μm for smFISH and 30 μm for IF) were cut using a cryostat (Leica, #CM3050S). The smFISH was performed using an RNAscope Fluorescent Multiplex Assay kit (Advanced Cell Diagnostics, ACD) following the manufacturer's instructions. Briefly, the brain slices were fixed with fixed with 4% PFA and washed with PBS, then dehydrated with ethanol and treated with hydrogen peroxide. The samples were then treated with the target retrieval reagents and protease for 10 min, after which the samples were hybridized with the probes at 40 °C for 2 h. Then the signals were amplified using the reagents included in the kit, followed by the labeling of relative channels with the TSA fluorophores. The samples were treated with the included blocking buffer before proceeding to DAPI staining and imaging. For the combined smFISH with CTB retrograde tracing, the brain samples were prepared from the mice with CTB injected into the relevant regions. The samples were treated following the manufacturer's instructions as described above till the labeling with the TSA fluorophore without being treated with the blocking buffer. The samples were then directly stained with DAPI, and the images were captured immediately. Probes (ACD) used were as followed: Foxp2 (Cat. No. 428791; Cat. No. 428791-C2); Slc17a7 (Cat. No. 416631-C3); Gad1 (Cat. No. 400951-C2); Ctgf (Cat. No. 314541); Etv1 (Cat. No. 557891-C3); Oprk1 (Cat. No. 316111; Cat. No. 316111-C2); Tshz2 (Cat. No. 431061-C3); Pou3f1 (Cat. No. 436421-C2); Chrna4 (Cat. No. 429871-C3); Chrna7 (Cat. No. 465161-C2); Chrnb2 (Cat. No. 449231-C2); Chrm1 (Cat. No. 495291-C2); Chrm2 (Cat. No. 495311).

For IF staining, the brain slices were washed in PBS twice for 10 min, followed by incubation in blocking buffer (5% goat serum, 5% BSA, and 0.1% Triton X-100 in PBS) for 40 min at room temperature. The samples were then incubated at 4 °C for 24 h with the primary antibody diluted in PBS containing 1% BSA. After the incubation, the samples were washed three times with washing buffer (0.1% Tween-20 in PBS) and incubated with the Alexa Fluor-conjugated secondary antibodies for 2 h at room temperature. The sections were washed three times, mounted, and imaged using a Zeiss LSM800 confocal microscope or Keyence BZ-X810 microscope. Antibodies used were as follows: rabbit anti-cFos (1:1000, Synaptic Systems, #226003), rabbit anti-ChAT (1:1000, Millipore, #ZRB1012-25UL), Donkey anti-rabbit Alexa Fluor 488 (Thermo Scientific, #A21206), Donkey anti-rabbit Alexa Fluor 568 (Thermo Scientific, # A10042).

### Stereotaxic surgeries
All stereotaxic surgeries were performed using a small-animal stereotaxic instrument (David Kopf Instruments, model 940) under general anesthesia by isoflurane (0.8 l min⁻¹; isoflurane concentration 1.5%) in oxygen. A feedback heater was used to maintain the body temperature of the mice. The eyes of the mice were kept moist using ophthalmic ointment throughout the surgery. A small craniotomy above the target brain region was performed using a dental drill. The viruses and CTB were injected into the target regions using a glass micropipette (10–20 μm in diameter at the tip) that was connected to a nanoliter injector (Nanoject III, Drummond Scientific, #3-200-207). For each site, 120 nl of viruses were injected at a flow rate of 1 nl s⁻¹ unless otherwise specified, allowing an additional 5 min for viral particles to diffuse before the pipette was slowly withdrawn. After the withdrawal, the wound was sutured, and the mice were allowed to recover in a warm blanket before being transferred to the housing cages. The mice were treated with meloxicam (Covetrus, Cat. No. 049756) subcutaneously daily for 3 days and monitored for another 2 days.

The coordinates of viral injection and implantation sites are based on previous literature and The Mouse Brain in Stereotaxic Coordinates (third edition) relative to the bregma (anterior-posterior, mediolateral, dorsoventral axis in mm): mPFC (1.94, 0.45, −2.50), PTN (−0.46, 0.30, −3.70), MD (−1.22, 0.50, −3.20), VM (−1.34, 0.75, −4.20), PAG (−4.75, 0.55, −2.70), NAc (+1.20, 0.60, 4.50), BLA (−1.34, 2.90, −4.50), HDB (−0.10, 1.45, −5.50).

## Chemogenetic and optogenetic manipulations

For chemogenetic manipulations of the mPFC *Foxp2*[+] neurons, AAV5-hSyn-DIO-mCherry (Addgene, Cat# 50459), AAV5-hSyn-DIO-hM4D(Gi)-mCherry (Addgene, Cat# 44362), or AAV5-hSyn-DIO-hM3D(Gq)-mCherry (Addgene, Cat# 44361) was bilaterally injected into the mPFC of *Foxp2*-Cre mice. The behavioral assays were performed 3 weeks later, and the mice were intraperitoneally injected with CNO (Cayman, Cat# 16882) at 1 mg/kg (for activation) or 5 mg/kg (for inactivation) in saline 20 min before each behavioral test as previously described[66,67]. For persistent inactivation of the mPFC *Foxp2*[+] neurons, rAAV2/9-hSyn-DIO-EYFP (BrainVTA, Cat# PT4250) or rAAV2/9-hSyn-DIO-TeNT-EYFP (BrainVTA, Cat# PT26697) was bilaterally injected into the mPFC of *Foxp2*-Cre mice, and behavioral assays were performed 5 weeks later.

For optogenetic activation of the terminals of mPFC *Foxp2*[+] neurons, rAAV2/9-EF1a-DIO-EYFP (BrainVTA, Cat# PT0012) or rAAV2/9-EF1A-DIO-ChR2-EYFP (BrainVTA, Cat# PT0001) was bilaterally injected into the mPFC of *Foxp2*-Cre mice, followed by the bilateral implantation of optical cannulas (diameter: 200 μm; N.A., 0.37; length, 4.0 mm; Inper Inc., China) 100 μm above the PTN/MD at a 20-degree angle or VM at a 10-degree angle, and secured with dental cement (Parkell, no. S380). For optogenetic activation of the cholinergic inputs to mPFC, the same viruses were injected into the HDB region of *ChAT*-Cre mice, and the optical cannulas were implanted above the mPFC. For all optogenetic manipulation, the mice were connected to the optical fiber for acclimation for 3 days before the behavioral tests and the light pulses (473 nm, 1.5 mW at the tip of the fiber, 20 Hz, 5-ms pulse) at 10-s ON/OFF cycle were generated by a 473-nm laser (OEM laser/OptoEngine) and used throughout the behavioral tests controlled by a waveform generator (Keysight).

## Neuronal tracing

For CTB retrograde tracing, three injections of CTB-555 (200 nl each) (Invitrogen, Cat# C34776) targeting the medial thalamic nuclei were performed to efficiently label the mPFC neurons projecting to the thalamus. For other regions, including mPFC, NAc, or amygdala, 120 nl of CTB-555 was unilaterally injected in individual mice, respectively. The brain tissues were collected 1 week later for smFISH. For specific axon labeling of the mPFC *Foxp2*[+] neurons, pAAV1-hSyn-DIO-mGFP-Synaptophysin-mRuby (Addgene, Cat# 71760) was unilaterally injected into the mPFC of *Foxp2*-Cre mice.

For monosynaptic retrograde rabies tracing, a 150 nl mixture of AAV helpers (rAAV2/9-EF1a-DIO-mCherry-TVA and rAAV2/9-EF1a-DIO-RVG) (BrainVTA, Cat# PT0023 and PT0207) was unilaterally injected into the mPFC of the *Foxp2*-Cre mice. Four weeks later, the same mice were injected with 250 nl of rabies virus (RV-ENVA-ΔG-EGFP) (BrainVTA, Cat# R01001) to the same location. The brain tissues were collected 7 days later for imaging and examination.

## Intracranial drug delivery

For intracranial drug delivery into the mPFC, the double cannula (Guide cannula: M3.5, C.C 1.0 mm, C = 2.5 mm; Injector: G1 = 0.5 mm; Dummy cannula: G2 = 0, mates with M3.5; RWD Life Science) with dust caps targeting both sides was directly implanted 0.5 mm above the mPFC and was secured with dental cement. Two weeks later, an injector (G2 = 0, mates with M3.5; RWD Life Science) filled with mineral oil and ABT-594 (Cayman Chemical, Cat. No. 22822) was connected to a pump (RWD Life Science) to slowly infuse the drug into both sides of mPFC (200 nl/site, 5 min) and 5 min were allowed for drug diffusion before the injector was slowly removed. The behavioral tests were performed another 5 min later. For smFISH of *cfos*, the mice were sacrificed 30 min after the drug injection. For intracranial drug delivery combined with chemogenetic inhibition, AAV5-hSyn-DIO-mCherry or AAV5-hSyn-DIO-hM4D(Gi)-mCherry was bilaterally injected into the mPFC, followed by the implantation of the double cannula. Three weeks later, the behavioral tests were performed 20 min after CNO injection (i.p.) and 5 min after ABT-594 infusion (intracranially).

## In vivo miniscopic calcium imaging and data processing

To probe the activity of the mPFC *Foxp2*[+] neurons in response to pain, in vivo miniscopic calcium imaging was performed as previously described[67,68]. Firstly, AAV1-hSyn-DIO-jGCaMP7s (Addgene, Cat# 104491) was unilaterally injected into the mPFC of the *Foxp2*-Cre mice. Two weeks later, the mice were anesthetized and fixed on a stereotaxic frame. The skull above the target region was carefully removed using a dental drilling and the brain tissue above the target region was aspirated using a 27-gauge needle with a blunt tip. Then an integrated GRIN lens (diameter 1.0 mm, length 4.0 mm; Inscopix) connected to a data acquisition system for monitoring the calcium signal was slowly lowered (~100 μm/min) down to 300 μm above the target region. The GRIN lens was further inserted to obtain an optimal field of view to image calcium signals, and then the GRIN lens was secured to the mice's skull with dental cement attached to a baseplate cover for protection. Saline was continuously added to avoid bleeding or drying of the tissue during the surgery. After the surgery, the mice were individually housed for 2 weeks before being used in imaging experiments.

Prior to calcium imaging recording, the mice were connected to the microscope through the head-fixed baseplate and acclimated to the handling of the experimenter and the testing chamber for 30 min each day for 3 days. On the day of data acquisition, the mice were connected to the microscope and allowed to move freely in the testing chamber. After 10 min of acclimation, the calcium images were obtained for 20–30 min using the nVoke system (Inscopix) at 20 Hz with 0.6–1.2 mW of LED power and 1.5 of gain. For individual mice, the imaging parameters were kept consistent across days.

For tail pinch stimulation, after acclimation, the mice were placed in the testing chamber without stimulation for 2 min and then an alligator clip was applied to the tail for 1 min, after which the clip was removed, and the mice were allowed to recover for 9 min, and the calcium signals were recorded during the entire process. For formalin stimulation, after acclimation the mice were recorded in the testing chamber without stimulation for 5 min, and then 20 μl of formalin was injected into the plantar of the mice, after which the mice were recorded for another 30 min in the chamber. In one cohort (Fig. 2), the mice were subjected to pinch stimulation, CFA treatment, and, 4 weeks later, formalin treatment. In another cohort (Supplementary Fig. 3f–i), the mice were subjected to formalin treatment only.

To detect the spontaneous activity of the neurons, the mice were acclimated to the microscope and testing environment with fixed brightness, temperature, and noise level one hour before the recordings. To avoid the potential effects of the light-dark cycle on the neuronal activity, all the recordings were performed in the afternoon and at the fixed time points for each individual animal. After the acclimation, the calcium activities were recorded for 15 min for the baseline. Then the mice were subjected to CFA injection under isoflurane anesthesia. One or seven days after the injection, the calcium activities were recorded again at the same time point after acclimation without any additional stimulation.

For data processing, the calcium traces were extracted and analyzed using the Inscopix Data Processing Software (version 1.9, Inscopix). The raw images were cropped, temporally downsampled (2×), spatially downsampled (2×), and further spatially filtered as well as motion-corrected using the default settings. The signals of putative individual cells were identified using an extended constrained non-negative matrix factorization (CNMF-E) algorithm and were normalized to $\Delta F$ over noise (settings: cell diameter: 8 pixels; minimum pixel correlation: 0.85; minimum peak-to-noise ratio: 15). The identified individual cells were then manually checked to be included based on their anatomy (size, shape, location), signal-to-noise ratio as well as

overlap in spatial location and signal with other cells. The calcium signal traces were further deconvoluted using OASIS (online active set method to infer spikes) methods to detect the potential calcium transients with the default noise ranges and spike signal-to-noise ratio of 8.0. Across the different days of recording for chronic pain, we first analyzed the calcium images to identify individual neurons in each recording and compared the spontaneous calcium activities of all the identified neurons among different days. Then the identified individual neurons were longitudinally registered with a minimum correlation of 0.9, and the matched neurons were further manually checked to be the same neurons across different days based on the locations and other visible landmarks (such as blood vessels).

To identify the neurons with significant change in the firing rate in response to tail pinch stimulation, the calcium traces were extracted and deconvoluted using OASIS to identify the calcium transients. We pooled together the identified transients of 2 min before and 1 min during the tail pinch stimulation, and used a permutation method to create 10,000 shuffled distributions of the transients. Then the shuffled transient change was calculated by subtracting the shuffled 0–2 min from that of the 3rd min. The neurons with actual transient change ranked higher than 99.5th percentile of the shuffled transient change were classified as activity-increased neurons, and those ranked lower than 0.5th percentile were classified as activity-decreased neurons. For the formalin test, the calcium signals of −5 min to 0 min and those of 0 min to 5 min were compared. For CFA-induced chronic inflammatory pain, the calcium signals of the baseline, CFA D1, and CFA D7 were compared using the same method.

### Behavioral assays

**von Frey test.** The mechanical sensory threshold of the mice was measured with a series of von Frey filaments (ranging from 0.04 to 2.0 g, Stoelting) using the Up-Down method as previously described[69]. Briefly, the mice were placed in a plastic chamber with a metal mesh floor and allowed to move freely. After 20 min of acclimation of the mice, the von Frey filaments were applied to the mid-plantar surface of the hind paw through the mesh floor and kept for 2 s, or until there was a positive response (paw withdrawal, flinching, licking, or shaking). To avoid the potential false positive responses caused by movement, the measurement was conducted again if a positive response was followed by other voluntary behaviors, such as locomotion, exploratory, or grooming behaviors.

**Hot plate test.** The hot plate test was used to measure thermal nociception of the mice. After being acclimated in the testing room for 20 min, the mice were placed on a hot plate (Ugo Basile) set at 52 °C, and the latencies to flinching or licking of hindpaws were measured. All animals were tested for 3 times with a minimum interval of 30 min. To avoid tissue injury, a cutoff latency was set at 30 s for the test.

**Formalin test.** Formalin test was performed as previously described[42]. Briefly, 20 µl of 2% formalin (Sigma-Aldrich, HT5011-1CS) was injected subcutaneously into the plantar surface of one hind paw of the mice with a 30-gauge syringe as quickly as possible. The licking behaviors were measured for 45 min starting from the injection of formalin, and the licking duration was calculated in 5-min bins. The licking duration of the mice during phase 1 (defined as 0–5 min after formalin injection) and phase 2 (defined as 15–45 min after formalin injection) was analyzed.

**Complete Freund's adjuvant (CFA) injection.** For CFA-induced inflammatory pain, 15 µl of CFA (Sigma, F5881-10ML) was subcutaneously injected into the plantar surface of the right hind paw of the mice under brief isoflurane anesthesia.

**Conditioned place preference.** The conditioned place preference test was used to measure the negative affect of pain and the effects of pain relief as previously described[67]. The testing apparatus consists of an infrared photobeam detector for tracking the mice's trajectories and two compartments, one of which is with rod style floor and the other has a grid-style floor (Med Associates). For chemogenetic manipulations, individual mice were placed in the chamber and allowed to freely explore the entire apparatus for 30 min (pretest) on day 1. On days 2–4, mice injected with saline were placed in the compartment with a grid floor (unconditioned compartment) for 30 min in the morning, while mice injected with CNO were placed in the opposite compartment with a rod floor (conditioned compartment) of the same apparatus for 30 min in the afternoon (with a minimum interval of 6 h between the two conditioning). On day 5, like day 1, the mice were placed in the chamber and allowed to freely explore the entire apparatus for 30 min (posttest). For optogenetic activation, the pretest and posttest on the first and last day were similarly performed. On days 2–4, the mice were connected with an optical fiber without light stimulation and placed in the compartment with a grid floor for 30 min in the morning, and in the afternoon, the mice were connected with the optical fiber with light stimulation as described above, and placed in the compartment with a rod floor for 30 min. For ABT-594 treatment, the pretest and posttest on the first and last day were similarly performed. On days 2–4, the mice were intracranially injected with ACSF and placed in the compartment with a grid floor for 30 min in the morning, and in the afternoon, the mice were injected with ABT-594 and placed in another compartment for 30 min. To calculate the CPP score, the time that mice spent in the conditioned compartment during pretest was subtracted from the time they spent during posttest.

### Statistics

The mice were randomly assigned to the groups, and the investigators were blind to the allocation of the groups or the outcome assessment. No statistical method was used to predetermine the sample size, but our sample sizes are similar to those reported in previous publications. Mice that, after histological inspection, had the location of the viral injection (reporter protein), cannula implantation, or the optic fiber(s) outside the area of interest were excluded. The data are presented as means ± SEM. Statistical analyses were performed using GraphPad Prism (version 8.0). Student's $t$-test and one-way or two-way RM ANOVA test with Tukey's or Sidak's *post hoc* multiple comparisons were applied to determine statistical differences. Statistical significance was set at $*P < 0.05$, $**P < 0.01$, $***P < 0.001$.

### Reporting summary

Further information on research design is available in the Nature Portfolio Reporting Summary linked to this article.

## Data availability

All data necessary to evaluate the conclusions of this study are available in the main text or the Supplementary Information files. Source data are provided with this paper.

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

## Acknowledgements

We thank the support of the Mouse Behavior Core of Harvard Medical School and its director, Dr. Barbara Caldarone. We thank Dr. Zhengdong Zhao for the help in using in vivo miniscopic calcium recording. This project was supported by the Open Philanthropy Foundation, the National Institutes of Health (1R01DA050589), and the HHMI. Y.Z. is an investigator of the Howard Hughes Medical Institute. This article is subject to HHMI's Open Access to Publications policy. HHMI lab heads have previously granted a nonexclusive CC BY 4.0 license to the public and a sublicensable license to HHMI in their research articles. Pursuant to those licenses, the author-accepted manuscript of this article can be made freely available under a CC BY 4.0 license immediately upon publication.

## Author contributions

Y.Z. conceived the project; G.X. designed the experiments; G.X. and Y.L. performed the experiments; X.Q. and G.X. analyzed the calcium imaging results; A.B. acquired the *Foxp2*-Cre mouse line and conducted preliminary staining and projection mapping; C.Z. analyzed the transcriptomic dataset. All authors are involved in data interpretation. G.X. and Y.Z. wrote the manuscript with input from all authors.

## Competing interests

The authors declare no competing interests.
