## [Transparent Peer Review file · Nature Communications]

A molecularly defined basalo-prefrontal-thalamic circuit regulates sensory and affective dimensions of pain in male mice

Corresponding Author: Professor Yi Zhang

Version 0:

Reviewer comments:

Reviewer #1

(Remarks to the Author)

Prefrontal cortex (PFC) is a well-known brain region to govern pain states and the layer 5 of mPFC has been well established to encode pain signaling. This manuscript identifies deactivation of specific Foxp2 neurons in the layer 6 of mPFC to regulate sensory pain and affective pain by projection to different subregions of thalamus. Furthermore, they traced cholinergic neurons from HDB of basal forebrain as up-stream inputs to mPFC and activation of $\alpha 4\beta 2$ nAChR in the mPFC is analgesic. Therefore, they revealed that the HDBChAT→mPFCFoxp2→thalamic nuclei circuit is crucial for pain modulation.

Overall, the manuscript provides some insights and extends the understanding of mPFC in pain regulation in the field. The manuscript is well written, however, some of the experiments are not rigorous designed which weakens the conclusion. I have some major and minor concerns to be addressed before considering publishing.

Major concerns:

1, For Fig 1a, the authors claim that around 63.2% of Foxp2+ neurons co-stained with CTB555, however, it seems non-CTB Foxp2+ neurons are more than the CTB-labelled neurons in the image. The authors may show more representative images to support this.

2, Similarly in Fig 1c, there are a lot of Foxp2+ neurons not labelling with Slc17a7 and it is hard to believe that the quantification is more than 90% as the authors claimed.

3, From Fig 2c, it seems the pinch, formalin test and CFA were performed in the same batch of mouse? Do the authors perform CFA first and then formalin model? How many mice were used per group? These should be clearly stated in the main text for figure legends.

4, In figure 2, the deactivation of Foxp2+ neurons is robust in the pinch stimulation, however, this deactivation is a bit low in formalin model (30%), with 17.66% increased and 51.4% no change, not like what the authors claimed that "a large subset was decreased" in line 193-197. So, it is hard to say what the real effect of these neurons are. Similarly, the deactivation is a bit low (25.8% and 43.5%) in the CFA model. The authors should use other methods like c-Fos staining or e-physics recording to confirm these results.

In addition, what causes the huge difference of deactivation of Foxp2+ neurons between the pinch and inflammatory pain models? The authors seem to ignore the increased activation and no changes neurons which take more amount than the deactivated one. What does the increased activation of neurons do?

5, In fig 3g, why the activation of Foxp2+ neurons were tested on day 2 and day 3 after CFA injection rather than day 1 and day 7 with deactivation of Foxp2+ neurons shown in Fig 2p-r. Do the authors confirm that these neurons were really activated by CNO in the CFA model? The quality of image is poor/over-exposed in Fig S4e and it is hard to see the signals. The author should quantify that virus-expressed c-Fos to confirm these neurons are activated rather than c-Fos only. Also, it is better to quantify the number normalized to sample size rather than the number only.

6, Fig 4a/b: Do the authors implant fibers with angle? There is no description in methods.

7, There are much more monosynaptic input neurons from IC, VM and BLA to the PLPFC than HDB in Fig. 6b and Fig. S5a, b. Do the authors check any of them to see whether these circuits impact nociceptive responses? Does HDB-PLPFC circuit influence affective pain? Basal ganglia and striatum also express cholinergic neurons. Do the authors checked these brain regions having inputs to mPFC?

8, In Fig6f, why the mechanical and thermal pain was tested on different day?

9, In Fig7j, the track is far away from the virus expression, the drug effect is questionable. The authors should provide

evidence to show that ABT-594 can activate Foxp2+ neurons in the mPFC by e-phys recording or at least c-Fos staining.

Minor:

- 1, For all behavioral responses with mechanical and thermal stimuli, like Fig 3e, 3g, Fig 4d/e/f, Fig 5b/c/d and etc, the Y axis should be fully described with Paw withdrawal threshold and Paw withdrawal latency.
- 2, No scale bar in Fig S3f.
- 3, Line 264: c-Fos missed a dash
- 4, The two phases of formalin model are 0-5min and 15-45min, the dash square in Fig3f, Fig 4g/h/i is not accurate.
- 5, In the manuscript, the authors use PLmPFC and mPFC in different place. Please keep consistent.
- 6, Fig 7k, the Y axis should be Paw withdrawal latency for HPT.

Reviewer #2

(Remarks to the Author)

Reviewer #3

(Remarks to the Author)

In the article titled "A molecularly defined basalo-prefrontal-thalamic circuit regulates sensory and affective dimensions of pain", Xie et al. report that a population of Foxp2-expressing neurons in mPFC layer 6 corticothalamic neurons. These neurons exhibit suppressed activity during pain states, and their specific activation significantly alleviates pain symptoms, including hyperalgesia and affective pain components. Furthermore, using viral tracing and immunostaining, the authors demonstrated that Foxp2 neurons project to distinct downstream thalamic nuclei to independently regulate sensory and affective pain dimensions. Finally, they revealed that these Foxp2 neurons receive inputs from HDBChAT neurons; activation via $\alpha 4\beta 2$ nAChRs induces analgesia, thereby delineating an HDBChAT \rightarrow mPFC Foxp2 \rightarrow thalamus neural circuit governing distinct pain components. Overall, this study demonstrates strong innovation and completeness, supported by robust experimental evidence and rigorous conclusions. However, there are several concerns with the findings that need to be addressed before consideration for publication.

Major Points:

1. Following CFA injection to induce inflammatory pain in figure 2C, the authors did not disclose whether formalin testing was performed on the same cohort of mice. Prior pain experiences—even after full recovery—could confound conclusions from subsequent formalin imaging recordings (such as pain memory effects).
2. The rationale for inhibiting Foxp2 neurons specifically in formalin-induced pain model mice is unclear (Figure S4). Since Foxp2 neuronal activity is already suppressed in pain states, chemogenetic inhibition in this model is functionally uninformative. This experiment should instead be conducted in naive mice.
3. The authors found Foxp2 primarily co-localized with L6 glutamatergic neurons. They should discuss how mPFC L6 glutamatergic neurons projecting to different thalamic nuclei (PTN, MD, VM) regulate distinct pain dimensions, citing existing literature. This comparison is essential to establish the specific regulatory role of Foxp2+ neurons versus broader L6 glutamatergic populations in pain processing.
4. This paper reported three key findings: 1) Deactivated activity of mPFC L6 Foxp2 neurons in pain states; 2) Activation of Foxp2 \rightarrow thalamus projections alleviates pain phenotypes; 3) HDBChAT \rightarrow mPFC Foxp2 inputs induce analgesia via $\alpha 4\beta 2$ nAChRs. Critical unresolved questions weaken the mechanistic narrative: How does initial pain signaling suppress mPFC Foxp2 activity? Under what conditions does the HDBChAT \rightarrow mPFC Foxp2 circuit exert analgesia? How is HDBChAT neuronal activity altered during pain? For example, does this circuit function as an endogenous analgesic system during CFA-induced pain recovery? If inhibited, would pain persistence be prolonged?

Minor Points:

1. CFA-induced pain (7 days) is inappropriately labeled "chronic", even Lines 302–305: two and three days after CFA injection... in chronic inflammatory pain. It is clearly inappropriate to refer to such a short period of time as chronic pain, please revise these descriptions.
2. Provide complete details in all figure legends, now sample size is not clarification. For example, Fig 2D: "n=280" clarify if this represents cells and specify the number of mice.
3. Explicitly label mice as CFA model mice in the figure/legend, such as Figure 3H.
4. For formalin licking duration (Phase I/II), use two-way ANOVA to demonstrate statistical significance.
5. Lines 371–375, Ambiguous description of nAChR subtype-encoding genes. Clarify.

Reviewer #4

(Remarks to the Author)

This study revealed new insight into the role of mPFC in regulating different dimensions of pain. They found that Foxp2 expression marks a group of layer 6 mPFC neurons that project to several medial thalamic nuclei and are deactivated by acute noxious stimuli and by chronic inflammatory conditions. Loss- and gain-of functional studies showed that inactivation of these neurons are crucial for driving nocifensive reactions, coping responses and affective pain. Interestingly, these distinct functions are apparently mediated via Foxp2+ neurons that project to distinct thalamic nuclei. Finally, they show that cholinergic inputs to Foxp2+ neurons, via activation of the $\alpha 4\beta 2$ nicotinic acetylcholine receptors, can modulate nociception, but less on affective pain. Most conclusions are well supported by the data and the finding is interesting, with

potential therapeutic impact.

I have only a few comments:

TO suggest that cholinergic signaling at mPFC has therapeutic potential, they need to assess the impact on affective pain in response to activation of cholinergic terminals from HDB brain regions, which had not been done in Figure 7. The CPP data described in Extended data Fig 7 did not reveal a robust impact on affective pain. Do the authors need to try different doses of the drug?

Version 1:

Reviewer comments:

Reviewer #1

(Remarks to the Author)

The all authors addressed most of my concerns and the paper is significantly improved. I have no more comments.

Reviewer #3

(Remarks to the Author)

The authors have addressed my concerns and the manuscript has been substantially improved. No more questions.

Reviewer #4

(Remarks to the Author)

Tha authors have carefully addressed the issues I raised. While the new CPP studies suggest the involvement in coping licking responses, but not affective pain and therefore reduced the potential clinical impact, the overall finding is still interesting.

Responses to Reviewers' Comments

We thank the reviewers for their critiques and helpful suggestions. To address their questions, we have performed a series of experiments and tried our best to improve the clarity of the manuscript. Our point-by-point response to the reviewers' questions are below:

Reviewer #1:

Prefrontal cortex (PFC) is a well-known brain region to govern pain states and the layer 5 of mPFC has been well established to encode pain signaling. This manuscript identifies deactivation of specific *Foxp2* neurons in the layer 6 of mPFC to regulate sensory pain and affective pain by projection to different subregions of thalamus. Furthermore, they traced cholinergic neurons from HDB of basal forebrain as up-stream inputs to mPFC and activation of $\alpha4\beta2$ nAChR in the mPFC is analgesic. Therefore, they revealed that the HDB^{ChAT}→mPFC^{*Foxp2*}→thalamic nuclei circuit is crucial for pain modulation. Overall, the manuscript provides some insights and extends the understanding of mPFC in pain regulation in the field. The manuscript is well written, however, some of the experiments are not rigorous designed which weakens the conclusion. I have some major and minor concerns to be addressed before considering publishing.

Response: We thank the reviewer for nicely summarizing our study and recognizing the novelty and significance of investigating the mPFC layer 6 *Foxp2*⁺ neurons in pain regulation. We have carefully considered all the comments and have revised the manuscript accordingly to ensure scientific rigor. We address the specific comments point-by-point below:

Major concerns:

1. For Fig 1a, the authors claim that around 63.2% of *Foxp2*⁺ neurons co-stained with CTB555, however, it seems non-CTB *Foxp2*⁺ neurons are more than the CTB-labelled neurons in the image. The authors may show more representative images to support this.

Response: We thank the reviewer for raising this question. The thalamus is a large structure that cannot be fully covered by a single CTB555 injection (120 nl) as described in the original manuscript. The proportion of *Foxp2*⁺ neurons co-labelled with CTB555 could therefore be influenced by the diffusion range and tracing efficiency of the tracer. To address this limitation, we repeated the experiment using three CTB555 injections (200 nl each) into the medial thalamic nuclei, where mPFC targets, to enhance the labeling efficiency. Quantification of the new results show that approximately 90% of *Foxp2*⁺ neurons in the mPFC are co-labelled with CTB555. Notably, about 90% of the CTB555 labelled cells co-express *Foxp2*, consistent with our results in the original manuscript, confirming our main conclusion that the *Foxp2*⁺ neurons constitute the major mPFC projection to the thalamus. These new data are presented in **Fig. 1a** of the revised manuscript as well as **Fig. R1** below for the reviewer's convenience.

Fig. R1. *Foxp2* marks the neurons in mPFC that project to thalamus

a, Diagram of the CTB injection into the thalamus of wildtype mouse. **b**, Representative image of the injection sites of CTB in the medial thalamus. **c**, Left, representative image of the mPFC with CTB labelled cells and smFISH showing the expression of *Foxp2*; middle and right, split channels of the mPFC for CTB (middle) and *Foxp2* (right). Scale bar, 200 μ m. **d**, Magnified image of the indicated region in **c** with colocalization of CTB and *Foxp2* (left), and the split channels for CTB (middle) or *Foxp2* (right). Scale bar, 200 μ m. **e**, Percentage of the *Foxp2*⁺ or *Foxp2*⁻ cells of all CTB⁺ cells (top) and percentage of the CTB⁺ or CTB⁻ cells of all *Foxp2*⁺ cells in the mPFC.

2. Similarly in Fig 1c, there are a lot of *Foxp2*⁺ neurons not labelling with *Slc17a7* and it is hard to believe that the quantification is more than 90% as the authors claimed.

Response: We thank the reviewer for raising this concern. To address this, we repeated the experiment and adjusted the imaging parameters so that the *Slc17a7*⁺ cells can be more effectively visualized. Representative images of the entire mPFC, along with higher-magnification views and split channels, are now provided to clearly illustrate the expression patterns of *Foxp2*, *Slc17a7* and *Gad1*. Quantification of the results show that over 95% of *Foxp2*⁺ cells in mPFC co-express *Slc17a7*, indicating that the mPFC *Foxp2*⁺ cells are predominantly glutamatergic neurons, consistent with our previous studies (PMID: 31519873; 37845544). The new data are now presented in **Fig. 1c** of the revised manuscript and also presented below in **Fig. R2** for the reviewer's convenience.

Fig. R2. The mPFC *Foxp2*⁺ cells are predominantly glutamatergic neurons

a, Representative image of the smFISH for *Foxp2* (red), *Slc17a7* (green) and *Gad1* (white) in the mPFC (left) and the split channels for *Foxp2* (red), *Slc17a7* (green) and *Gad1* (white) respectively. Scale bar, 200 μ m. **b**, Magnified image of the indicated region in (white square) and the split channels for *Foxp2* (red), *Slc17a7* (green) and *Gad1* (white), respectively. Scale bar, 20 μ m. **c**, Percentage of the *Foxp2*⁺ cells co-expressing *Slc17a7* or *Gad1* in the mPFC.

3. From Fig 2c, it seems the pinch, formalin test and CFA were performed in the same batch of mouse? Do the authors perform CFA first and then formalin model? How many mice were used per group? These should be clearly stated in the main text for figure legends.

Response: We appreciate the reviewer for the helpful comment. Five mice were used for miniscopic calcium imaging, and all the five mice underwent pinch, formalin and CFA treatments on the indicated days after surgery (**Fig 2c**). As previous studies reported that CFA-induced inflammatory pain persists for 7 days stably and gradually resolves within 2-3 weeks (PMID: 29623029; 26179626), we first applied the CFA treatment to all the five mice. Four weeks later, after full recovery from the CFA-induced inflammatory pain, the same mice were subjected to formalin treatment. In addition, we also conducted the miniscopic calcium imaging for formalin treatment in a new cohort of mice without prior pinch or CFA treatment in the revised manuscript (**Extended Data Fig. 3f-i**). The details have been clarified in the figure legend of the revised manuscript as suggested.

4. In figure 2, the deactivation of *Foxp2*⁺ neurons is robust in the pinch stimulation, however, this deactivation is a bit low in formalin model (30%), with 17.66% increased and 51.4% no change, not like what the authors claimed that “a large subset was decreased” in line 193-197. So, it is hard to say what the real effect of these neurons are. Similarly, the deactivation is a bit low (25.8% and 43.5%) in the CFA model. The authors should use other methods like c-Fos staining or e-phys recording to confirm these results.

In addition, what causes the huge difference of deactivation of *Foxp2*⁺ neurons between the pinch and inflammatory pain models? The authors seem to ignore the increased activation and no

changes neurons which take more amount than the deactivated one. What does the increased activation of neurons do?

Response: We thank the reviewer for these insightful questions, which give us an opportunity to clarify our analysis. To examine the activity of the mPFC *Foxp2*⁺ neurons in response to noxious stimulation, we averaged the calcium event frequency of all recorded *Foxp2*⁺ neurons and compared it before and after stimulation. The results showed significant decreases in the overall calcium events of *Foxp2*⁺ neurons consistently in response to pinch, formalin, and CFA treatments compared to the baseline (**Fig. 2f, j and m** in the revised manuscript), indicating consistent deactivation of this neuronal population under the three pain conditions. To improve clarity, the orders of the **Fig. 2f, g, j and k** are re-organized, and the descriptions have also been updated in the revised manuscript.

Despite this significant overall deactivation, we also observed heterogeneity among the mPFC *Foxp2*⁺ neurons, with different subpopulations showing increased or decreased activity following noxious stimulation. To quantify these subpopulations, the calcium events from baseline and post-stimulation periods were shuffled 10,000 times to generate a shuffled distribution. Neurons with actual calcium transient changes exceeding 99.5th percentile of the shuffled distribution were classified as neurons with “increase” in calcium activity, and those below the 0.5th percentile as “decrease”, as described previously (PMID: 30655440). To avoid confusion, we now label the neurons not reaching these statistical thresholds as “not significant” rather than “no change” in the revised manuscript (**Fig. 2g, k**), since these neurons also contribute to the overall decrease in the activity of *Foxp2*⁺ neuronal population.

We thank the reviewer for the suggestion of using c-Fos staining. c-Fos as an immediate-early gene is highly expressed in activated neurons and is widely used as a neuronal activity marker. Although the overall activity of the mPFC *Foxp2*⁺ neurons decreases in response to noxious stimulation, a subset of these neurons shows increased activity compared to baseline. This heterogeneity could elevate c-Fos expression and confound the interpretation of overall *Foxp2*⁺ neuronal activity. Electrophysiological recording was not available to us and, moreover, would not capture short-term dynamics of *Foxp2*⁺ neuronal activity in response to acute noxious stimulation or the population-wide activity of *Foxp2*⁺ neurons. Instead, we used miniscopic calcium imaging to monitor the calcium dynamics of hundreds of individual *Foxp2*⁺ neurons in vivo. Despite variation in the proportion of neurons reaching statistical significance across experiments, our results consistently show decreased overall activity of the *Foxp2*⁺ neuronal population in response to pinch, formalin, and CFA treatment (**Fig. 2f, j and m**).

Regarding the differences in deactivation levels observed in the pinch and inflammatory pain models, the effect was more robust in the pinch condition (**Fig. 2f, j and m**), with a greater number of neurons classified as significantly deactivated (**Fig. 2g, q and r**). Three factors may account for this difference: (1) the intensity and modality of pain differ between the two models; (2) the valence of pain varies, with pinch representing an urgent salient threat that evokes tonic coping behaviors such as licking and escape; and (3) the duration of pain differs, as prolonged inflammation may lead to adaption and activation of the endogenous analgesic pathways that mitigate the deactivation of *Foxp2*⁺ neurons. Despite these differences, the overall calcium event frequency of *Foxp2*⁺ neurons is significantly decreased in both models (**Fig. 2f, m**), supporting

our conclusion that the mPFC *Foxp2*⁺ neurons are deactivated in both acute mechanical and prolonged inflammatory pain.

To investigate the roles of the activated *Foxp2*⁺ neurons, we performed chemogenetic inactivation under pain conditions (**Extended Data Fig. 4**). Because the overall activity of the mPFC *Foxp2*⁺ neurons has already been decreased during acute or CFA-induced pain, this manipulation primarily targeted the activated subset. However, the inactivation produced no significant effects on nocifensive thresholds, coping behaviors or negative affect of pain (**Extended Data Fig. 4**), suggesting that the activated *Foxp2*⁺ neurons contribute minimally to sensory or affective pain, or that their effects are outweighed by the population-wide deactivation under pain conditions. Given the critical roles of activating the *Foxp2*⁺ neuronal population in regulating multiple aspects of pain, we hypothesize that the pain-activated *Foxp2*⁺ neurons may help maintain cortical output to the thalamus and are essential for initiating the endogenous pain relief pathways following initial pain signaling.

5. In fig 3g, why the activation of *Foxp2*⁺ neurons were tested on day 2 and day 3 after CFA injection rather than day 1 and day 7 with deactivation of *Foxp2*⁺ neurons shown in Fig 2p-r. Do the authors confirm that these neurons were really activated by CNO in the CFA model? The quality of image is poor/over-exposed in Fig S4e and it is hard to see the signals. The author should quantify that virus-expressed c-Fos to confirm these neurons are activated rather than c-Fos only. Also, it is better to quantify the number normalized to sample size rather than the number only.

Response: We thank the reviewer for the questions. As noted above, CFA-induced inflammatory pain typically persists for 7 days stably and gradually resolves in 2 to 3 weeks (PMID: 29623029; 26179626). Accordingly, we conducted all behavioral assays within 9 days after CFA treatment. Prior to the behavioral tests, the mice were acclimated to the testing room and testing apparatus. Behavioral assays were performed on separate days to minimize stress and avoid potential confounding effects between experiments. Specifically, the von Frey test and hot plate test were performed on days 2 and 3, respectively, and the conditioned place preference test was conducted from days 5 to 9 after CFA treatment.

To verify the effectiveness of chemogenetic manipulation, we repeated immunostaining of c-Fos administration and quantified c-Fos⁺ cells as suggested. Compared with the mCherry control group, our results show that the proportion of virus-expressing neurons co-expressing c-Fos is significantly lower in the hM4Di group and higher in the hM3Dq group, confirming that CNO treatment effectively inhibited or activated the targeted neurons, respectively. In addition, we also observed increased c-Fos expression in neurons that did not express the virus in the hM3Dq group. It has been reported that the mPFC and thalamus are reciprocally connected and that the mPFC can drive activation of the thalamocortical neurons (PMID: 29628187). It is therefore possible that activation of the mPFC *Foxp2*⁺ neurons enhances cortical output to the thalamus and, through strengthened thalamic feedback, modulates the activity of other mPFC neurons. The new data is now presented in **Extended Data Fig. 4e** in the revised manuscript and **Fig. R3** below for the reviewer's reference.

Fig. R3. c-Fos staining for validation of chemogenetic manipulation

a, Representative images of the c-Fos (green) and mCherry (red) of the mPFC after CNO treatment of mice injected AAV expressing mCherry (left), hM4Di-mCherry (middle) or hM3Dq-mCherry (right). Scale bars, 20 μ m. **b**, Percentage of the AAV-expressing cells co-expressing c-Fos in the mPFC of the three groups.

6. Fig 4a/b: Do the authors implant fibers with angle? There is no description in methods.

Response: We thank the reviewer for raising this question. To achieve bilateral optogenetic manipulation, the optical fibers were implanted above the PTN or MD at a 20-degree angle, and above the VM at a 10-degree angle. We have updated this information in methods in the revised manuscript (line 661-662).

7. There are much more monosynaptic input neurons from IC, VM and BLA to the PLPFC than HDB in Fig. 6b and Fig. S5a, b. Do the authors check any of them to see whether these circuits impact nociceptive responses? Does HDB-PLPFC circuit influence affective pain? Basal ganglia and striatum also express cholinergic neurons. Do the authors checked these brain regions having inputs to mPFC?

Response: We thank the reviewer for the thoughtful comment. Cholinergic signaling has long been implicated in pain processing (PMID: 28890048), and previous studies have shown that the mPFC is strongly modulated by cholinergic signaling (PMID: 21836018), which is severely impaired in chronic pain (PMID: 28137966). In line with these findings, our results showed that the $\alpha 4\beta 2^*$ nAChR, an acetylcholine receptor with known antinociceptive effects (PMID: 9417028), is highly enriched in *Foxp2*⁺ neurons in the mPFC (Fig. 7a-f). Thus, the current study focused on the cholinergic inputs from the HDB and their translational potential in pain modulation, rather than other afferent pathways. We appreciate the reviewer's point that the mPFC *Foxp2*⁺ neurons receive strong monosynaptic inputs from IC, VM and BLA, all of which are also involved in pain processing. We agree with the reviewer and believe that these circuits likely influence the *Foxp2*⁺ neuronal activity and therefore contribute to nociceptive regulation. However, as our current study centers on the cholinergic pathway and already includes seven main figures and several extended data figures, we consider investigation of the

other inputs to the *Foxp2*⁺ neurons in pain processing to be an important direction for future studies.

To address the reviewer's question regarding the role of the HDB-mPFC circuit in affective pain, which was also raised by other reviewers, we activated the terminals of HDB cholinergic neurons in the mPFC (as in **Fig. 6d**) and conducted conditioned place preference (CPP) test. We observed no significant difference in the time spent in the chamber or CPP scores between the control and ChR2 group (**Fig. 6h** in the revised manuscript, or **Fig. R4b,c** below), indicating that activation of the HDB cholinergic inputs in mPFC does not markedly alleviate the affective component of CFA-induced inflammatory pain. The mice were also subjected to formalin test to measure the coping behaviors, and we found that the licking duration is significantly decreased during phase 2 in the ChR2 group (**Fig. 6f** in the revised manuscript, or **Fig. R4d,e** below), suggesting that activation of the HDB^{ChAT}-mPFC circuit significantly reduces coping behaviors in mice. The new data are now presented in **Fig. 6** of the revised manuscript and **Fig. R4** below for the reviewer's reference.

Fig. R4. Effects of HDB^{ChAT}-mPFC circuit in affective pain and coping behaviors

a, Diagram of surgery on *ChAT*-Cre mice with virus injection into HDB and optical fiber implanted above the mPFC. **b**, Diagram of the CPP test after CFA treatment coupled with optogenetic manipulation. **c**, Time that the mice spent in the chamber coupled with optogenetic activation before and after CPP training (left), and the CPP scores of the mice (right). **d**, Diagram of formalin test. **e**, Licking duration of the mice in response to formalin treatment (left) and the summarized licking duration of the mice in phase 1 (middle) and phase 2 (right) of the formalin test.

As suggested by the reviewer, we re-examined the samples of rabies tracing shown in **Fig. 6a**. No detectable RV-GFP labelled cells were observed in the caudate putamen (CPu), globus pallidus (GP) subthalamic nucleus (STh) or reticular part of the substantia nigra (SNr). These results further confirm that the cholinergic inputs to the mPFC *Foxp2*⁺ neurons originate

primarily from the basal forebrain, particularly the HDB. The new data are presented in **Fig. R5** below.

Fig. R5. Rabies virus tracing of the mPFC *Foxp2*⁺ neurons showing the basal ganglion

a, Representative image of caudate putamen (CPu) and globus pallidus (GP). **b**, Representative image of the subthalamic nucleus (STh). **c**, Representative image of the substantia nigra (SNr). Scale bars, 200 μ m. 3V, third ventricle; cp, cerebral peduncle; HYT, hypothalamus; LV, lateral ventricle; MB, mammillary body.

8. In Fig6f, why the mechanical and thermal pain was tested on different day?

Response: We thank the reviewer for the question. As described above and in the methods, the mice were acclimated to the testing room and testing apparatus before the behavioral measurement. To minimize stress to the mice and avoid potential confounding effects between the experiments, all behavioral assays were performed on separate days. In addition, the experimental time points were consistent with other parts of the study, including the chemogenetic experiments, in which the pharmacokinetics of CNO may only be sufficient for one behavioral assay per administration (PMID: 26889809). Therefore, the mechanical and thermal pain were measured on different days throughout the study.

9. In Fig7j, the track is far away from the virus expression, the drug effect is questionable. The authors should provide evidence to show that ABT-594 can activate *Foxp2*⁺ neurons in the mPFC by e-phys recording or at least c-Fos staining.

Response: We appreciate the reviewer for raising this concern. As suggested, we performed smFISH for *Foxp2* and *cfos* after intracranial injection of ABT-594. Our results show that the proportion of the *Foxp2*⁺ neurons co-expressing *cfos* is significantly increased after ABT-594 treatment compared to control group (ACSF treatment), confirming that ABT-594 can activate the *Foxp2*⁺ neurons in the mPFC. We also noticed an increase in the number of *cfos*⁺*Foxp2*⁻ cells in mPFC following ABT-594 treatment. It is likely due to activation of the mPFC *Foxp2*⁺ neurons engaging in the thalamocortical feedback loop, thereby modulating the activity of other mPFC neurons, consistent with our findings from chemogenetic manipulation of the *Foxp2*⁺ neurons (**Fig. R3**). The new data is presented in the **Extended Data Fig. 7b** in the revised manuscript, and **Fig. R6** below for the reviewer's reference.

Fig. R6. smFISH for *Foxp2* and *cfos* after intracranial ACSF or ABT-594 treatment

a. Representative images of the smFISH for *Foxp2* (red) and *cfos* (green) of the mPFC after intracranial ACSF (left) or ABT-594 (right) treatment. Scale bar, 100 μ m. **b.** Percentage of the *Foxp2*⁺ cells co-expressing *cfos* in the mPFC of the two groups.

Minor concern:

1. For all behavioral responses with mechanical and thermal stimuli, like Fig 3e, 3g, Fig 4d/e/f, Fig 5b/c/d and etc, the Y axis should be fully described with Paw withdrawal threshold and Paw withdrawal latency.

Response: We thank the reviewer for the comment and we have updated all the relevant figures as suggested.

2. No scale bar in Fig S3f.

Response: We thank the reviewer for pointing this out, and the scale bar has been added in the revised manuscript. We also checked the scale bars for all the figures of the revised manuscript and corrected some mistakes in the legends of **Fig. 1**.

3. Line 264: c-Fos missed a dash

Response: We thank the reviewer for pointing this out. We have carefully read and tried our best to correct all typos in the revised manuscript.

4. The two phases of formalin model are 0-5min and 15-45min, the dash square in Fig3f, Fig 4g/h/i is not accurate.

Response: We thank the reviewer for pointing this out. To improve clarity, we have removed the dashed square and explicitly define the time periods for phase 1 and phase 2 in the figures.

5. In the manuscript, the authors use PLmPFC and mPFC in different place. Please keep consistent.

Response: We thank the reviewer for pointing out this and we have now used mPFC to describe the region in the revised manuscript.

6. Fig 7k, the Y axis should be Paw withdrawal latency for HPT.

Response: We apologize for the oversight, and the error has been corrected in the revised manuscript.

Reviewer #2:

Response: We thank the reviewer for the effort and hope our responses above have addressed all of your concerns.

Reviewer #3:

In the article titled “A molecularly defined basalo-prefrontal-thalamic circuit regulates sensory and affective dimensions of pain”, Xie et al. report that a population of *Foxp2*-expressing neurons in mPFC layer 6 corticothalamic neurons. These neurons exhibit suppressed activity during pain states, and their specific activation significantly alleviates pain symptoms, including hyperalgesia and affective pain components. Furthermore, using viral tracing and immunostaining, the authors demonstrated that *Foxp2* neurons project to distinct downstream thalamic nuclei to independently regulate sensory and affective pain dimensions. Finally, they revealed that these *Foxp2* neurons receive inputs from HDB^{ChAT} neurons; activation via $\alpha 4\beta 2$ nAChRs induces analgesia, thereby delineating an HDB^{ChAT}→mPFC^{*Foxp2*}→thalamus neural circuit governing distinct pain components. Overall, this study demonstrates strong innovation and completeness, supported by robust experimental evidence and rigorous conclusions. However, there are several concerns with the findings that need to be addressed before consideration for publication.

Response: We appreciate the reviewer for recognizing the novelty, completeness and scientific robustness of our study on the HDB^{ChAT}→mPFC^{*Foxp2*}→thalamus neural circuit in governing distinct pain components. We address the reviewer’s specific comments point-by-point below:

Major points:

1. Following CFA injection to induce inflammatory pain in figure 2C, the authors did not disclose whether formalin testing was performed on the same cohort of mice. Prior pain experiences—even after full recovery—could confound conclusions from subsequent formalin imaging recordings (such as pain memory effects).

Response: We thank the reviewer for the comment. Five mice were used for the miniscopic calcium imaging, and all the five mice were subjected to the pinch, CFA and formalin treatment on the indicated days after surgery (as shown in **Fig. 2c**). Previous studies have reported that CFA-induced inflammatory pain persists stably for about 7 days and gradually resolves within 2 to 3 weeks (PMID: 29623029; 26179626). In the original manuscript, we first performed CFA treatment, and four weeks later after the mice recovered from the CFA-induced inflammatory pain, the same mice were treated by formalin. We have now clarified these experimental details in the figure legends, results, and methods sections of the revised manuscript.

We also appreciate the reviewer for raising the important concern regarding the potential confounding effects of prior pain experiences on subsequent formalin testing. We agree with the reviewer and to address the concern, we performed a new batch of miniscopic calcium imaging. Three mice were used in the new experiment and subjected to formalin treatment directly after recovery from surgery and acclimation, without prior pinch or CFA treatment. Consistent with our previous findings (**Fig. 2h-k**), the new results show a significant decrease in calcium event frequency during both phase 1 and phase 2 (**Extended Data Fig. 3f-i** in the revised manuscript, and **Fig. R7** below). These findings further support our conclusion that the mPFC *Foxp2*⁺ neurons are deactivated in both phases following formalin treatment. The new data are presented in **Extended Data Fig. 3f-i** of the revised manuscript and also in **Fig. R7** below for the reviewer's reference.

Fig. R7. The mPFC *Foxp2*⁺ neurons of naïve mice are deactivated in response to formalin treatment

a, Timeline of the treatment (top) and diagram of calcium recording for formalin treatment (bottom). **b**, Left, heatmap of the calcium event frequency averaged to 1 min of bin of all recorded neurons before and after formalin treatment ($n = 274$ neurons from 3 mice). The white dotted line indicates the time of formalin injection. Right, representative traces showing fluorescence intensity change ($\Delta F/F$) of individual neurons before and after formalin treatment. The red and the yellow backgrounds indicate phase 1 and phase 2 of formalin test, respectively. **c**, Cumulative distribution (left) and the average (right) of calcium event frequency in all identified neurons during baseline (BL), phase 1 and phase 2. $n = 274$ neurons from 3 mice. **d**, Percentages of the neurons with significantly decrease (red), increase (yellow), or not significant (n.s.) change (gray) in calcium event frequency during phase 1 compared to baseline.

2. The rationale for inhibiting *Foxp2* neurons specifically in formalin-induced pain model mice is unclear (Figure S4). Since *Foxp2* neuronal activity is already suppressed in pain states, chemogenetic inhibition in this model is functionally uninformative. This experiment should instead be conducted in naïve mice.

Response: We thank the reviewer for raising this question. In the miniscopic calcium imaging experiments (**Fig. 2** and **Extended Data Fig. 3**), although the overall activity of the mPFC *Foxp2*⁺

neurons was decreased, a subset was activated by noxious stimulation. To investigate their functional roles, we examined the effects of chemogenetic inhibition of the mPFC *Foxp2*⁺ neurons on different aspects of pain (**Extended Data Fig. 4**). Since the *Foxp2*⁺ neuronal population is already deactivated, the chemogenetic inhibition primarily targeted the activated subset. However, no significant changes in sensory or affective pain were observed, suggesting that this subset contributes minimally to pain processing or that its effects are masked by the population-wide suppression under pain conditions. We have clarified the description and interpretation in the revised manuscript (line 265-272).

3. The authors found *Foxp2* primarily co-localized with L6 glutamatergic neurons. They should discuss how mPFC L6 glutamatergic neurons projecting to different thalamic nuclei (PTN, MD, VM) regulate distinct pain dimensions, citing existing literature. This comparison is essential to establish the specific regulatory role of *Foxp2*⁺ neurons versus broader L6 glutamatergic populations in pain processing.

Response: We appreciate the reviewer for this thoughtful suggestion. The mPFC L6 glutamatergic neurons are mainly consisted of two populations: the L6 CT (corticothalamic) and L6 IT (intratelencephalic) neurons (PMID: 33972100). The mPFC *Foxp2*⁺ neurons are predominantly L6 CT neurons and constitute the major output of the mPFC to the thalamus (**Fig. 1** and **Extended Data Fig. 1**). In contrast, the L6 IT neurons do not project to the thalamus but instead target intratelencephalic regions, including the contralateral mPFC and nucleus accumbens (NAc) (as indicated in **Extended Data Fig. 1c-e**). It has been reported that layer 5 pyramidal tract (L5 PT) neurons also project to the thalamus, albeit to a lesser extent (PMID: 36550291), which is consistent with our observation that approximately 10% of the cells retrogradely labelled from thalamus do not express *Foxp2* (**Fig. 1a**). Both the L5 PT and L6 CT neurons project to the mediodorsal (MD) and ventromedial (VM) thalamic nuclei, with L6 CT neurons acting as modulators that evoke robust firing of thalamocortical neurons through facilitating synapses (PMID: 29628187). Outputs from the mPFC to other brain regions such as NAc, amygdala, and periaqueductal gray (PAG) arise primarily from IT and PT neurons, respectively (PMID: 33972100). L5 PT projection to the PAG has been well described in pain regulation through descending modulatory pathways (PMID: 31501573), whereas IT neurons projecting to the NAc or amygdala modulate the affective dimensions of pain and associated comorbidities (PMID: 30150924; 36917193). Our studies identify a critical role of the L6 CT neurons, marked by *Foxp2*, in regulating both sensory and affective aspects of pain via discrete thalamic projections, thereby filling in an important gap in understanding the contribution of mPFC L6 neurons to pain processing. We thank the reviewer for the comment and have updated the discussion section accordingly in the revised manuscript (line 495-507).

4. This paper reported three key findings: 1) Deactivated activity of mPFC L6 *Foxp2* neurons in pain states; 2) Activation of *Foxp2*→thalamus projections alleviates pain phenotypes; 3) HDB^{ChAT}→mPFC^{*Foxp2*} inputs induce analgesia via $\alpha 4\beta 2$ nAChRs. Critical unresolved questions weaken the mechanistic narrative: How does initial pain signaling suppress mPFC *Foxp2* activity? Under what conditions does the HDB^{ChAT}→mPFC^{*Foxp2*} circuit exert analgesia? How is HDB^{ChAT} neuronal activity altered during pain? For example, does this circuit function as an endogenous analgesic system during CFA-induced pain recovery? If inhibited, would pain persistence be prolonged?

Response: We appreciate the reviewer for the insightful comments. As summarized by the reviewer, our study focuses on a molecularly defined cell type in the mPFC, the L6 CT neurons marked by *Foxp2*, which constitute the major output of mPFC to thalamus. We demonstrated that these neurons are deactivated under pain conditions and that their activation relieves both sensory and affective aspects of pain. Additionally, we delineated the circuit-specific functions of these neurons in modulating different aspects of pain through projections to the PTN, MD, and VM thalamic nuclei. Furthermore, we identified the cholinergic inputs from the HDB as functionally critical regulators of mPFC *Foxp2*⁺ neurons and highlighted the therapeutic potential of targeting $\alpha 4\beta 2$ nicotinic acetylcholine receptor for pain management.

By comprehensively characterizing this molecularly defined cell type, its projection-specific functions, and its modulation by cholinergic signaling, our study already has seven main figures and multiple extended data figures. We agree that the reviewer has raised some unresolved valuable questions for understanding the mechanisms of the HDB^{ChAT}→mPFC^{*Foxp2*}→thalamus circuit in pain processing and its role in the endogenous analgesic system. Considering the scope and findings of the current work, we view these remaining questions belong to a follow-up study.

Minor points:

1. CFA-induced pain (7 days) is inappropriately labeled “chronic”, even Lines 302–305: two and three days after CFA injection... in chronic inflammatory pain. It is clearly inappropriate to refer to such a short period of time as chronic pain, please revise these descriptions.

Response: We thank the reviewer for pointing this out. We have modified the relevant descriptions to CFA-induced inflammatory pain in the revised manuscript.

2. Provide complete details in all figure legends, now sample size is not clarification. For example, Fig 2D: “n=280” clarify if this represents cells and specify the number of mice.

Response: We thank the reviewer for pointing this out and we have now clarified the details in the figure legends in the revised manuscript.

3. Explicitly label mice as CFA model mice in the figure/legend, such as Figure 3H.

Response: We thank the reviewer for the comment. We have labeled the mice with CFA treatment explicitly in both figures and figure legends in the revised manuscript.

4. For formalin licking duration (Phase I/II), use two-way ANOVA to demonstrate statistical significance.

Response: We thank the reviewer for the suggestion. We have performed two-way ANOVA for the statistical analysis and updated the statistical significances and conclusions in the revised manuscript.

5. Lines 371–375, Ambiguous description of nAChR subtype-encoding genes. Clarify.

Response: We thank the reviewer for raising this concern. The description has been updated to improve clarity in the revised manuscript.

Reviewer #4:

This study revealed new insight into the role of mPFC in regulating different dimensions of pain. They found that *Foxp2* expression marks a group of layer 6 mPFC neurons that project to several medial thalamic nuclei and are deactivated by acute noxious stimuli and by chronic inflammatory conditions. Loss- and gain-of functional studies showed that inactivation of these neurons are crucial for driving nocifensive reactions, coping responses and affective pain. Interestingly, these distinct functions are apparently mediated via *Foxp2*⁺ neurons that project to distinct thalamic nuclei. Finally, they show that cholinergic inputs to *Foxp2*⁺ neurons, via activation of the $\alpha 4\beta 2$ nicotinic acetylcholine receptors, can modulate nociception, but less on affective pain. Most conclusions are well supported by the data and the finding is interesting, with potential therapeutic impact.

Response: We thank the reviewer for nicely summarizing our study and the positive comments regarding the rigor, novelty and significance of the study. We address this reviewer's specific comments below:

1. To suggest that cholinergic signaling at mPFC has therapeutic potential, they need to assess the impact on affective pain in response to activation of cholinergic terminals from HDB brain regions, which had not been done in Figure 7.

Response: We appreciate the reviewer for pointing this out and for the helpful suggestion. Following the suggestion, we performed optogenetic activation of the HDB cholinergic terminals in the mPFC, as described in **Fig. 6d**, and conducted CPP experiment to assess the effects on affective pain. Surprisingly, we did not observe significant differences in either the time spent in the chamber or the CPP scores between the control and ChR2 group (**Fig. 6h** in the revised manuscript, or **Fig. R4b,c** above), indicating that activation of the HDB cholinergic terminals in the mPFC does not significantly alleviate the negative affect of CFA-induced inflammatory pain. We also performed the formalin test to assess coping behaviors and found that the licking duration during phase 2 was significantly reduced in the ChR2 group (**Fig. 6f** in the revised manuscript, or **Fig. R4d,e** above), suggesting that activation of the HDB^{ChAT}-mPFC circuit significantly reduces the coping behaviors in mice. Our results indicate that the HDB cholinergic projection to the mPFC regulates the nociceptive sensitivity and coping behaviors, but not the negative affect of pain. The new data are presented in **Fig. 6h,f** in the revised manuscript and **Fig. R4** above for the reviewer's convenience.

2. The CPP data described in Extended data Fig 7 did not reveal a robust impact on affective pain. Do the authors need to try different doses of the drug?

Response: We thank the reviewer for the comment. In the original manuscript, the mice received ABT-594 treatment at a dosage of 5 pmol per site for the CPP experiment. Since no significant effects on affective pain were observed, we repeated the experiment with a new cohort of mice using a higher dosage of ABT-594 (25 pmol per site), which is sufficient to produce antinociceptive effects in von Frey test (**Extended Data Fig. 7a** in the revised manuscript) and is within the higher range reported in the literature (PMID: 9417028). Consistent with the previous results, we again found no significant difference in the CPP test between the control and ABT-594 treatment groups, indicating that ABT-594 treatment does not effectively alleviate the affective dimension of pain. The new data are presented in **Extended Data Fig. 7e** in the revised manuscript and **Fig. R8** below for the reviewer's convenience.

Fig. R8. No effects of intracranial ABT-594 treatment on affective pain even at a higher dosage

a, Diagram of CPP test for the mice with CFA-induced inflammatory pain paired with ACSF or ABT-594 treatment at a dosage of 25 pmol per site. **b**, The time spent in the chamber paired with ABT-594 treatment before and after the CPP training (left) and the CPP scores of the mice (right).

We believe that we have addressed the great majority of the concerns from the reviewers. We want to thank the reviewers for their valuable suggestions, as addressing their questions significantly improved the quality of our work. We hope that the reviewers are satisfied with our response and revision.

Sincerely,

Yi Zhang, Ph.D
HMS, BCH, HHMI